# DENSITY-SOFTMAX: EFFICIENT TEST-TIME MODEL FOR UNCERTAINTY ESTIMATION AND ROBUSTNESS UNDER DISTRIBUTION SHIFTS

## ABSTRACT

Sampling-based methods, e.g., Deep Ensembles and Bayesian Neural Nets have become promising approaches to improve the quality of uncertainty estimation and robust generalization. However, they suffer from a large model size and high latency at test-time, which limits the scalability needed for low-resource devices and real-time applications. To resolve these computational issues, we propose Density-Softmax, a sampling-free deterministic framework via combining a density function built on a Lipschitz-constrained feature extractor with the softmax layer. Theoretically, we show that our model is the solution of minimax uncertainty risk and is distance-aware on feature space, thus reducing the over-confidence of the standard softmax under distribution shifts. Empirically, our method achieves competitive results with state-of-the-art techniques in terms of uncertainty and robustness, while having a lower number of model parameters and a lower latency at test-time.

## 1 INTRODUCTION

The ability of models to produce high-quality uncertainty estimation and robustness is crucial for reliable DNN in high-stake applications (e.g., healthcare, finance, decision-making, etc.). In principle, a **reliable model** permits graceful failure, signaling when it is likely to be wrong (**uncertainty**), and also generalizes better under distribution shifts (**robustness**) (Tran et al., 2022). Additionally, in order to be widely used in real-world scenarios, the reliable DNN model also necessarily needs to be fast and lightweight (**efficiency**). This efficiency criterion can be considered in two phases, training-time and test-time. At **training-time**, an inefficient DNN model might be acceptable given the high computational resources in development. However, at **test-time**, inefficiency is a critical issue for users when the model needs to be deployed on low-resource devices and in real-time applications (Hinton et al., 2015; Nado et al., 2021).

Deterministic ERM (Vapnik, 1998) model nowadays can be efficient due to being **sampling-free**, i.e., it only needs a single forward pass with a single DNN model to produce the softmax probability. However, it often struggles with over-confidence and overfitting (Guo et al., 2017). This poor performance usually occurs when the test data is far and does not come from the same distribution as the training set (Ovadia et al., 2019; Minderer et al., 2021).

**Table 1:** Comparison in uncertainty (U), robustness (R), and test-time efficiency (E).

| Method | U | R | E |
|---|---|---|---|
| ERM | ✗ | ✗ | ✓ |
| Ensembles | ✓ | ✓ | ✗ |
| Ours | ✓ | ✓ | ✓ |

To improve uncertainty estimation and robustness under distribution shifts, recent **sampling-based** approaches (Collier et al., 2021; Dusenberry et al., 2020; Wen et al., 2020; 2018; Gal & Ghahramani, 2016b) have shown promising results. Among these works, the best performance in practice so far is based on Deep Ensembles (Lakshminarayanan et al., 2017; Hansen & Salamon, 1990; Nado et al., 2021). However, this approach suffers from a heavy computational burden as it requires more model parameters and multiple forward passes, leading to inefficiency at test-time. To tackle this challenge, sampling-free methods (Liu et al., 2020a; Havasi et al., 2021), and lightweight sampling-based models (Wen et al., 2020; Dusenberry et al., 2020) have been recently proposed. Nevertheless, besides generally performing worse than Deep Ensembles, these methods are also less computationally efficient than Deterministic ERM (Nado et al., 2021) (e.g., Tab. 1).

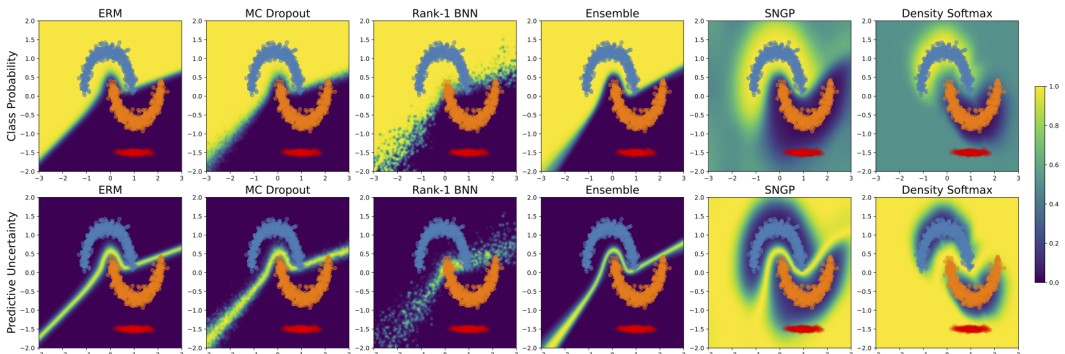

**Figure 1:** The class probability $p(y|x)$ (Top Row) and predictive uncertainty $var(y|x) = p(y|x) \times (1-p(y|x))$ surface (Bottom Row) as background colors in a comparison between our Density-Softmax and different approaches on the two moons 2D classification. Training data for positive (Orange) and negative classes (Blue) are shown. OOD data (Red) are not observed during training. **Our Density-Softmax achieves distance awareness with a uniform class probability and high uncertainty value on OOD data**. *A quick demo is available at https://colab.research.google.com/drive/1fdsAW_J4WKBFSTa3Hc2EDo_ZbjsoSle7?usp=sharing.*

Toward a model that keeps the Deep Ensembles performance with test-time efficiency similar to Deterministic ERM, we introduce Density-Softmax, a sampling-free single-model via a combination of a density function and a Lipschitz-constrained feature extractor. By using regularization to enforce the 1-Lipschitz constraint in training, our model can improve the robustness under distribution shifts. In addition, by combining the feature density function with the softmax layer via a single forward pass, our method only needs a small number of additional parameters and latency when compared to Deterministic ERM. Importantly, this combination helps Density-Softmax be feature distance aware, i.e., its associated uncertainty metrics are monotonic functions of feature distance metrics, leading to a good uncertainty notion of when the test feature is near or far from the training set. This distance-aware property is important to help DNN for both calibration and OOD detection, however, it is often not guaranteed for typical DNN models (Liu et al., 2020a) (e.g., Fig. 1).

To summarize, our model includes three main components: a Lipschitz-constrained feature extractor, a lightweight Normalizing-Flows density model on the feature space, and a classifier with the softmax layer. In training-time, the feature extractor is pre-trained using ERM objective and gradient-penalty regularization, aiming to archive 1-Lipschitz constraint. After that, the Normalizing-Flows model estimates a marginal density of the learned low-dimensional feature space. Finally, the classifier is fine-tuned with feature likelihood from the density model. In test-time, the feature likelihood is combined directly with the logit vector of the classifier to produce a softmax probability with only a single forward pass.

**Our contributions can be summarized as:**

1. We introduce Density-Softmax, a reliable, sampling-free, and single DNN framework via a direct combination of a density function built on a Lipschitz-constrained feature extractor with the softmax layer. Notably, our algorithm does not require any data augmentation (Hendrycks* et al., 2020; Zhang et al., 2018), the OOD set, or the post-hoc re-calibration technique in training.

2. We formally prove that our model is the solution to the minimax uncertainty risk, distance awareness on the feature space, and can reduce over-confidence of the standard softmax when the test feature is far from the training set.

3. We empirically show that our framework achieves a competitive robust generalization and uncertainty estimation performance with SOTA on the Toy dataset with ResFFN-12-128, shifted CIFAR-10-100 with Wide Resnet-28-10, and ImageNet with ResNet-50. Importantly, Density-Softmax requires only a single forward pass and a lightweight feature density function. Therefore it has fewer parameters and is much faster than other baselines at test-time.

## 2   RELATED WORK

**Uncertainty and Robustness.** Nado et al. (2021) has studied the uncertainty and robustness of modern deep learning approaches extensively in the benchmark of SOTA baselines, mainly evaluating NLL, in-out accuracy for robust generalization, and ECE for calibrated uncertainty. More modern discussion of reliable DNN can be found in the literature of Tran et al. (2022). Among these methods, sampling-based approaches are widely used, from Gaussian Process (Gardner et al., 2018; Lee et al., 2018), Dropout (Gal & Ghahramani, 2016a; Gal et al., 2017), BNN (Blundell et al.,

2015; Maddox et al., 2019), to the SOTA Ensembles (Lakshminarayanan et al., 2017). However, these methods often have a high number of model parameters. To resolve this issue, lightweight sampling-based models, e.g., BatchEnsemble (Wen et al., 2020), Rank-1 BNN (Dusenberry et al., 2020), and Heteroscedastic (Collier et al., 2021) have been proposed recently.

**Sampling-free methods.** To tackle the scalability challenge in sampling-based methods, novel sampling-free methods have been investigated, including replacing the loss function (Wei et al., 2022; Malinin & Gales, 2018; 2019; Kotelevskii et al., 2022; Karandikar et al., 2021), the output layer (Van Amersfoort et al., 2020; Mukhoti et al., 2023; Tagasovska & Lopez-Paz, 2019; Wang et al., 2022; Liu et al., 2020b), or computing a closed-form posterior in Bayesian inference (Kopetzki et al., 2021; Sensoy et al., 2018; Riquelme et al., 2018; Snoek et al., 2015; Kristiadi et al., 2020; Charpentier et al., 2022). A detailed comparison with NatPN (Charpentier et al., 2022) is in Apd. B.4. Nevertheless, these methods often only focus on improving uncertainty quality without improving accuracy (see Apd. B.4), or even using additional re-calibration set to enhance this performance (Mukhoti et al., 2023). In the scope of uncertainty and robustness (Nado et al., 2021), there exist distillation approaches (Vadera et al., 2020; Malinin et al., 2020) and closest to our work is MIMO (Havasi et al., 2021), Posterior Net (Charpentier et al., 2020), and SNGP (Liu et al., 2020a). Although these methods can improve efficiency at test-time, they still underperform Deep Ensembles in terms of reliability, and Deterministic ERM in terms of the model's test-time efficiency.

**Improving uncertainty quality via density estimation.** Enhancing uncertainty estimates via density function has shown promising results in practice (Kuleshov & Deshpande, 2022; Charpentier et al., 2020; Mukhoti et al., 2023; Kotelevskii et al., 2022). Theoretically, Charpentier et al. (2022) proves with a small number of parameters, the Normalizing-Flows model can improve the uncertainty notion of DNN by providing a high likelihood when the test feature is close and a low likelihood when that is far from training data. Yet, these work often customize the last layer of DNN with sensitive priors and only evaluate the uncertainty quality. In this work, our Density-Softmax utilizes a Normalizing-Flows model on the Lipschitz-constrained feature together with the logit vector of the classifier to improve the robust accuracy, uncertainty quality, and test-time efficiency.

**Improving robustness via 1-Lipschitz constraint.** 1-Lipschitz Neural Nets have been widely used to train certifiably robust classifiers in practice (Tsuzuku et al., 2018; Béthune et al., 2022; Li et al., 2019; Searcód, 2006). It is theoretically shown to be able to defend against adversarial attack (Li et al., 2019), preserve accuracy on IID, and improve the robust generalization on OOD data (Béthune et al., 2022). However, it is not clear whether this property can contribute to better uncertainty estimation. In this work, we integrate the Lipschitz-constrained feature extractor from gradient-penalty regularization (Gulrajani et al., 2017) with our density estimation component to achieve better robust accuracy, uncertainty quality, and test-time efficiency.

## 3 DENSITY-SOFTMAX

**Notation and Problem setting.** Let $\mathcal{X} \subset \mathbb{R}^{d_x}$ and $\mathcal{Y} \subset \mathbb{R}$ be the sample and label space. Denote the set of joint probability distributions on $\mathcal{X} \times \mathcal{Y}$ by $\mathcal{P}_{\mathcal{X} \times \mathcal{Y}}$. A dataset is defined by a joint distribution $\mathbb{P}(x, y) \in \mathcal{P}_{\mathcal{X} \times \mathcal{Y}}$, and let $\mathcal{P}$ be a measure on $\mathcal{P}_{\mathcal{X} \times \mathcal{Y}}$, i.e., whose realizations are distributions on $\mathcal{X} \times \mathcal{Y}$. Denote training data by $D_s = \{(x_i, y_i)\}_{i=1}^n$, where $n$ is the number of data points in $D_s$, i.e., $(x_i, y_i) \sim \mathbb{P}_s(x, y)$ and $\mathbb{P}_s(x, y) \sim \mathcal{P}$. In a standard learning setting, a model is trained on $D_s$ and arrives at a good generalization on the test set $D_t = \{(x_j, y_j)\}_{j=1}^m$, where $m$ is the number of data points in $D_t$, i.e., $(x_j, y_j) \sim \mathbb{P}_t(x, y)$ and $\mathbb{P}_t(x, y) \sim \mathcal{P}$. In the typical Independent-identically-distributed (IID) setting, $\mathbb{P}_t(x, y)$ is similar to $\mathbb{P}_s(x, y)$, and let us use $\mathbb{P}_{iid}(x, y)$ to represent the IID test distribution. In contrast, $\mathbb{P}_t(x, y)$ is different with $\mathbb{P}_s(x, y)$ if $D_t$ is Out-of-Distribution (OOD) data, and let us use $\mathbb{P}_{ood}(x, y)$ to represent the OOD test distribution.
In the classification setting of representation learning, we predict a target $y \in \mathcal{Y}$, where $\mathcal{Y}$ is discrete with $K$ possible categories by using a forecast $h = \sigma(g \circ f)$, which composites a feature extractor $f : \mathcal{X} \to \mathcal{Z}$, where $\mathcal{Z} \subset \mathbb{R}^{d_z}$ is feature space, a classifier embedding $g : \mathcal{Z} \to \mathbb{R}^K$, and a softmax layer $\sigma : \mathbb{R}^K \to \Delta_y$ which outputs a probability distribution $W(y) : \mathcal{Y} \to [0, 1]$ within the set $\Delta_y$ of distributions over $\mathcal{Y}$; the value of probability density function of $W$ is $w$.

**Motivation and Overview.** Toward a reliable and efficient test-time framework, we introduce Density-Softmax. Specifically, to improve uncertainty quantification, our idea is based on the solution of the **minimax uncertainty risk** (Meinke & Hein, 2020), i.e., $\inf_{\mathbb{P}(Y|X) \in \mathcal{P}} \left[ \sup_{\mathbb{P}^*(Y|X) \in \mathcal{P}_*} S(\mathbb{P}(Y|X), \mathbb{P}^*(Y|X)) \right]$ (1), where $S(., \mathbb{P}^*(Y|X))$ is strictly proper

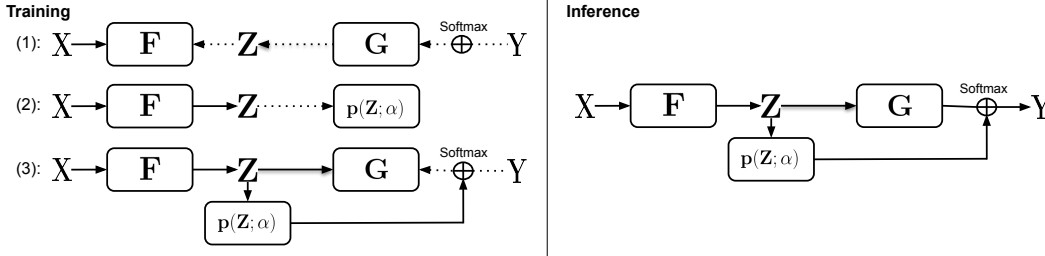

**Figure 2:** The overall architectures of Density-Softmax, including an encoder $f$, a classifer $g$, and a density function $p(Z; \alpha)$. The rectangle boxes represent these functions. The dashed lines represent the backward, solid lines represent the forward pass. The circle with two cross lines represents the softmax layer. The 3 training steps and inference process follow Algorithm 1.

scoring rules, $\mathbb{P}(Y|X)$ is predictive, and $\mathbb{P}^*(Y|X)$ is the data-generation distribution. When $\mathcal{X}_{ood} = \mathcal{X}/\mathcal{X}_{iid}$, for the Brier Score (Brier, 1950), the solution of the risk was shown by Liu et al. (2020a) as $\mathbb{P}(Y|X) = \mathbb{P}(Y|X_{iid})\mathbb{P}^*(X_{iid}) + \mathbb{U}(Y|X_{ood})\mathbb{P}^*(X_{ood})$, where $X_{iid}$ is IID, $X_{ood}$ is OOD sample variable, and $\mathbb{U}$ stands for uniform distribution. According to this result, we summarize the overview of Density-Softmax, a solution of the minimax uncertainty risk by Cor. 4.4, in Fig. 2. Under our framework, the predictive distribution is equivalent to $\mathbb{P}(Y|X) = \sigma(p(Z; \alpha) * g(Z))$, with $Z = f(X)$, where $f : \mathcal{X} \to \mathcal{Z}$ is the feature extractor, $g : \mathcal{Z} \to \mathbb{R}^K$ is the last classifier layer, $\sigma : \mathbb{R}^K \to \Delta_y$ is the softmax layer, and $p(Z; \alpha)$ is the density function to measure the representation distance on feature space $\mathcal{Z}$ with parameter $\alpha$.

To improve robustness and test-time efficiency, we use the gradient-penalty regularization (Gulrajani et al., 2017) to enforce the feature extractor $f$ to be 1-Lipschitz based on the Rademacher theorem:

**Theorem 3.1.** *(Rademacher, Theorem 3.1.6 (Federer, 1969)) If $f : \mathbb{R}^{d_x} \to \mathbb{R}^{d_z}$ is a locally Lipschitz continuous function, then $f$ is differentiable almost everywhere. Moreover, if $f$ is Lipschitz continuous, then $L(f) = \sup_{x \in \mathbb{R}^n} ||\nabla_x f(x)||_2$, where $L(f)$ is the Lipschitz constant of $f$.*

*Remark* 3.2. The gradient-penalty $(||\nabla_x f(x)||_2 - 1)^2$ enforces $\sup_{x \in \mathbb{R}^n} ||\nabla_x f(x)||_2 = 1$ and Thm. 3.1 suggests $f(x)$ satisfy 1-Lipstchiz constraint by $L(f) = 1$, i.e., $||f(x_1) - f(x_2)||_2 \le ||x_1 - x_2||_2$. This prevent $f(x)$ being overly sensitive to the meaningless perturbations of the sample and assures that if the sample is similar, the feature will be similar as well. This 1-Lipstchiz $f(x)$ is proved to be robust on the corrupted data by the Local Robustness Certificates property:

*Property* 3.3. **(Local Robustness Certificates).** *(Property 1 (Béthune et al., 2022; Tsuzuku et al., 2018)) For any 1-Lipschitz $f(x)$, i.e., $L(f) = 1$, the robustness radius $\epsilon$ of binary classifier $c = sign \circ f$ at example $x$ verifies $\epsilon \ge ||f(x)||$, where $\epsilon = \min_{\delta \in \mathbb{R}^{d_x}} ||\delta||$ s.t. $c(x + \delta) \ne c(x)$. ( This result can be generalized to the multi-class case by Béthune et al. (2022)).*

**Training.** Based on the aforementioned motivation, in the first training step, we optimize the model by using ERM (Vapnik, 1998) and gradient-penalty regularization (Gulrajani et al., 2017) from training data $D_s$ by solving

$$\min_{\theta_{g,f}} \left\{ \mathbb{E}_{(x,y) \sim D_s} \left[ -y \log \left( \sigma(g(f(x))) \right) + \lambda (||\nabla_x f(x)||_2 - 1)^2 \right] \right\}, \tag{2}$$

where $\theta_{g,f}$ is the parameter of encoder $f$ and classifier $g$, $\lambda$ is the gradient-penalty coefficient, and $||\nabla_x f(x)||_2$ is the Spectral norm of the Jacobian matrix $\nabla_x f(x)$.

After that, we freeze the parameter $\theta_f$ of $f$ to estimate density on the learned representation space $\mathcal{Z}$ by positing a Normalizing-Flows model for $p(Z; \alpha)$ (Papamakarios et al., 2021; Dinh et al., 2017), then fitting MLE to yield $\alpha$ and scale $p(Z; \alpha)$ to a specified range. We use Normalizing-Flows to fit the statistical density model $p(Z; \alpha)$ because it is simple, lightweight, provable, and provides exact log-likelihood (Charpentier et al., 2022). Specifically, we do MLE w.r.t. the logarithm by optimizing

$$\max_{\alpha} \left\{ \mathbb{E}_{x \sim D_s} \left[ \log(p(z; \alpha)) := \log(p(t; \alpha)) + \log \left| \det \left( \frac{\partial t}{\partial z} \right) \right| \right] \right\}, \text{ with } z = f(x), \tag{3}$$

where random variable $t = s_\alpha(f(x))$ and $s$ is a bijective differentiable function.

Finally, to enhance generalization after combining with the likelihood value of the density function $p(Z; \alpha)$, we update the weight of classifier $g$ to normalize with the likelihood value by optimizing with the objective function as follows

$$\min_{\theta_g} \left\{ \mathbb{E}_{(x,y) \sim D_s} \left[ -y \log \left( \sigma(p(z; \alpha) \times g(z)) \right) \right] \right\}, \text{ with } z = f(x). \tag{4}$$

---

**Algorithm 1** Density-Softmax: Training and Inference

---

**Training input:** Training data $D_s$, encoder $f$, density function $p(Z; \alpha)$, classifier $g$, learning rate $\eta$, gradient-penalty coefficient $\lambda$.
**for** $e = 1 \rightarrow$ pre-train epochs **do**
   Sample $D_B$ with a mini-batch $B$ for source data $D_s$
   $\theta_{g,f} \leftarrow \theta_{g,f} - \eta \nabla_{\theta_{g,f}} \mathbb{E}_{(x,y) \in D_B} \left[ -y \log\left(\sigma(g(f(x)))\right) + \lambda(\|\nabla_x f(x)\|_2 - 1)^2 \right]$
**end for**
**for** $e = 1 \rightarrow$ train-density epochs **do**
   Sample $D_B$ with a mini-batch $B$ for source data $D_s$
   $\alpha \leftarrow \alpha - \eta \nabla_\alpha \mathbb{E}_{z=f(x) \sim D_B} \left[ -\log(p(t; \alpha)) - \log\left| \det\left(\frac{\partial t}{\partial z}\right) \right| \right]$
**end for**
**Scale:** $p(Z; \alpha)$ to $(0, 1]$
**for** $e = 1 \rightarrow$ re-optimize classifier epochs **do**
   Sample $D_B$ with a mini-batch $B$ for source data $D_s$
   $\theta_g \leftarrow \theta_g - \eta \nabla_{\theta_g} \mathbb{E}_{(z=f(x),y) \in D_B} \left[ -y \log\left(\sigma(p(z; \alpha) * g(z))\right) \right]$
**end for**
**Inference (test) input:** Test sample $x_t$
$z_t = f(x_t); p(y = i|x_t) = \frac{\exp(p(z_t; \alpha) \times (z_t^\top \theta_{g_i}))}{\sum_{j=1}^K \exp(p(z_t; \alpha) \times (z_t^\top \theta_{g_j}))}, \forall i \in \mathcal{Y}$

---

**Inference.** After completing the training process, for a new test $x_t$ at the test-time, we perform prediction by combining the density function on feature space $p(z_t; \alpha)$ and classifier $g$ to make a prediction with only a single forward pass by the following formula

$$p(y = i|x_t) = \frac{\exp(p(z_t; \alpha) \times (z_t^\top \theta_{g_i}))}{\sum_{j=1}^K \exp(p(z_t; \alpha) \times (z_t^\top \theta_{g_j}))}, \forall i \in \mathcal{Y}, \tag{5}$$

where $z_t = f(x_t)$ is the feature presentation for test sample $x_t$.

*Remark* 3.4. (**Computational efficiency at test-time**) Eq. 5 shows that the complexity of Density-Softmax at test-time is similar to Deterministic ERM by requiring only a single forward pass to produce the softmax probability. Its computation only needs to additionally compute $p(z_t; \alpha)$, therefore, only higher than Deterministic ERM by the additional parameter $\alpha$ and the latency of $p(z_t; \alpha)$. These number is often very small in practice by the tables in Section 5.

The pseudo-code for the training and inference processes of our proposed Density-Softmax framework is presented in Algorithm 1. It is worth noticing that due to the likelihood $p(Z; \alpha) \in (-\infty; +\infty)$, the exponential function in Eq. 5 can return a $NaN$ if the likelihood $p(Z; \alpha) \rightarrow \infty$ in implementation. Therefore, we need to scale it to the range of $(0, 1]$ to avoid this numerical issue. The detail for controlling the scale of likelihood is provided in Appendix B.1.

## 4 THEORETICAL ANALYSIS OF UNCERTAINTY ESTIMATION

Regarding uncertainty, we first inherit the Feature Distance Awareness Def. (Liu et al., 2020a):

**Definition 4.1. (Feature distance awareness).** The predictive distribution $\sigma(g(z_t))$ on test feature $z_t = f(x_t)$ is said **feature distance aware** if there exists $u(z_t)$, a summary statistics of $\sigma(g(z_t))$, that quantifies model uncertainty (e.g., entropy, predictive variance, etc.) and reflects the distance between $z_t$ and the training data $Z_s$ w.r.t. $\|\cdot\|_{\mathcal{Z}}$, i.e., $u(z_t) := v(d(z_t, Z_s))$, where $v$ is a monotonic function and $d(z_t, Z_s) := \mathbb{E} \|z_t - Z_s\|_{\mathcal{Z}}$ is the distance between $z_t$ and the training data $Z_s$.

And we recall a Lemma when $p(Z; \alpha)$ is a Normalizing-Flows model (Papamakarios et al., 2021):

**Lemma 4.2.** *(Lemma 5 (Charpentier et al., 2022)) If $p(Z; \alpha)$ is parametrized with a Gaussian Mixture Model (GMM) or a radial Normalizing-Flows, then $\lim_{d(z_t, Z_s) \rightarrow \infty} p(z_t; \alpha) \rightarrow 0$.*

Lemma 4.2 intuitively leads Eq. 5 to reasonable uncertainty estimation for the two limit cases of strong IID and OOD data. In particular, for very unlikely OOD data, i.e., $d(z_t, Z_s) \rightarrow \infty$, the prediction will go to uniform. Conversely, for very likely IID data, i.e., $d(z_t, Z_s) \rightarrow 0$, the prediction follows the in-domain predictive distribution. We formally show this property below:

**Theorem 4.3.** *(**Uniform and in-domain prediction for strong OOD and IID data**). Density-Softmax provides a uniform prediction, i.e., $\sigma(p(z_{ood}; \alpha) * g(z_{ood})) = \mathbb{U}$ when $d(z_{ood}, Z_s) \rightarrow \infty$ by $p(z_{ood}); \alpha) \rightarrow 0$, and preserves the in-domain prediction, i.e., $\sigma(p(z_{iid}; \alpha) * g(z_{iid})) = \sigma(g(f(x_{iid})))$ when $d(z_{iid}, Z_s) \rightarrow 0$ by $p(z_{iid}); \alpha) \rightarrow 1$. (The proof is provided in Appendix A.1).*

Based on this theorem, we next show that Density-Softmax is the solution of the minimax uncertainty risk in Eq. equation 1 and satisfies Def. 4.1 about feature distance awareness:

**Corollary 4.4.** *(Solution of the minimax uncertainty risk).* *Density-Softmax's prediction is the optimal solution of the minimax uncertainty risk, i.e.,* $\sigma(p(f(X); \alpha) * g(f(X)) = \underset{\mathbb{P}(Y|X) \in \mathcal{P}}{\arg\inf} \left[ \sup_{\mathbb{P}^*(Y|X) \in \mathcal{P}*} S(\mathbb{P}(Y|X), \mathbb{P}^*(Y|X)) \right]$ *. (The proof is provided in Appendix A.2).*

**Proposition 4.5.** *(Distance aware on feature space).* *The predictive distribution of Density-Softmax* $\sigma(p(z = f(x); \alpha) * (g \circ f(x)))$ *is distance aware on the feature space* $\mathcal{Z}$.

*Proof sketch.* Prop. 4.5 is proved by showing (1) $p(z = f(x); \alpha)$ is monotonically decreasing w.r.t. distance $\mathbb{E} \|z_t - Z_s\|_{\mathcal{Z}}$ and (2) $p(z = f(x); \alpha) * g$ is distance aware. Full proof is in Apd. A.3. □

*Remark* 4.6. Prop. 4.5 shows our Density-Softmax is distance aware on the feature representation $\mathcal{Z}$, i.e., its predictive probability reflects the distance between the test feature and the training set. This is a necessary condition for a DNN to achieve high-quality uncertainty estimation (Liu et al., 2020a). By showing $p(Z; \alpha)$ is monotonically decreasing w.r.t. feature distance, this proves when the likelihood of $p(Z; \alpha)$ is high, our model is certain on IID data, and when the likelihood of $p(Z; \alpha)$ decreases on OOD data, the certainty will decrease correspondingly.

Following these results, we finally present Density-Softmax can enhance the uncertainty quality of DNN with the standard softmax by reducing its over-confidence in the proposition below:

**Proposition 4.7.** *(Reducing over-confidence of the standard softmax).* *If the predictive distribution of the standard softmax* $\sigma(g \circ f)$ *makes* $acc(B_m) \leq conf(B_m), \forall B_m$ *in Eq. 40, then Density-Softmax* $\sigma((p(f; \alpha) * g) \circ f)$ *can improve calibrated-uncertainty in terms of ECE (Eq. 40), i.e.,* $ECE(\sigma((p(f; \alpha) * g) \circ f)) \leq ECE(\sigma(g \circ f))$. *(The proof is provided in Appendix A.4).*

*Remark* 4.8. The condition $acc(B_m) \leq conf(B_m), \forall B_m$ is a specific case of the over-confidence for every $M$ bins. And if so, Prop. 4.7 shows Density-Softmax can reduce ECE of the standard softmax, which also be empirically confirmed in Fig. 3 in the next section.

## 5 EXPERIMENTS

### 5.1 EXPERIMENTAL SETTINGS

**Datasets.** We utilize 6 commonly used datasets under distributional shifts (Nado et al., 2021), including Toy dataset (Liu et al., 2020a) with two moons and two ovals to visualize uncertainty and clustering. CIFAR-10-C, CIFAR-100-C, and ImageNet-C (Hendrycks & Dietterich, 2019) for the main benchmarking. To evaluate real-world shifts, we experiment on SVHN (Netzer et al., 2011) and CIFAR-10.1 (Recht et al., 2018). The detail of each dataset is in Apd. C.1.

**Baseline.** For a fair comparison, we use the fair and high-quality baseline of Nado et al. (2021). This compares **10 recently SOTA methods** in reliability, including **sampling-free models**: Deterministic ERM (Vapnik, 1998), Posterior Net (Charpentier et al., 2020), SNGP (Liu et al., 2020a), MIMO (Havasi et al., 2021), **sampling-based models**: MC Dropout (Gal & Ghahramani, 2016b), MFVI BNN (Wen et al., 2018), Rank-1 BNN (Dusenberry et al., 2020), Heteroscedastic (Collier et al., 2021), and Ensembles families with BatchEnsemble (Wen et al., 2020) and Deep Ensembles (Lakshminarayanan et al., 2017). The detail of each method is in Apd. C.2.

**Evaluation Metrics.** To evaluate the generalization, we use NLL and Accuracy. For uncertainty estimation, we visualize uncertainty surfaces (entropy, different predictive variances), predictive entropy, AUPR/AUROC for OOD detection, and ECE with 15 bins for calibration. To compare the robustness under distributional shifts, we evaluate every OOD set in each dataset. To compare computational efficiency, we count the number of model parameters for storage requirements and measure latency in milliseconds per sample **at test-time**. The infrastructure detail is in Apd. C.3.

**Implementation.** We follow experimental settings based on the source code of Nado et al. (2021). We train models on the train set excluding data augmentation (Hendrycks* et al., 2020; Zhang et al., 2018), test on the original IID test, and aforementioned OOD sets. The performance is evaluated on backbones ResFFN-12-128 (Liu et al., 2020a) for the Toy, Wide Resnet-28-10 (Zagoruyko & Komodakis, 2016) for CIFAR-10-100, and Resnet-50 (He et al., 2016) for ImageNet. Implementation details are in Apd. C.3. All source code to reproduce results are available at *this anonymous GitHub*.

**Table 2:** Results for Wide Resnet-28-10 on CIFAR-10, averaged over 10 seeds: negative log-likelihood (lower is better), accuracy (higher is better), and expected calibration error (lower is better). NLL, Acc, and ECE represent performance on IID test set. cNLL, cAcc, and cECE are NLL, accuracy, and ECE averaged over OOD CIFAR-10-C's corruption types & intensities, oNLL, oAcc, and oECE are for real-world shift CIFAR-10.1. #Params is the number of model parameters, Latency is the milliseconds to inference per sample on RTX A5000, ours is colored by blue (lower are better). Best scores with the significant test are marked in **bold**. A more comparison with recent sampling-free methods is in Apd. B.4

| Method | NLL(↓) | Acc(↑) | ECE(↓) | cNLL(↓) | cAcc(↑) | cECE(↓) | oNLL(↓) | oAcc(↑) | oECE(↓) | #Params(↓) | Latency(↓) |
|---|---|---|---|---|---|---|---|---|---|---|---|
| ERM | 0.159 | 96.0 | 0.023 | 1.05 | 76.1 | 0.153 | 0.40 | 89.9 | 0.064 | **36.50M** | **518.12** |
| MC Dropout | 0.145 | 96.1 | 0.019 | 1.27 | 70.0 | 0.167 | 0.39 | 89.9 | 0.058 | 36.50M | 4,319.01 |
| MFVI BNN | 0.211 | 94.7 | 0.029 | 1.46 | 71.3 | 0.181 | 0.49 | 88.1 | 0.070 | 72.96M | 809.01 |
| Rank-1 BNN | 0.128 | 96.3 | **0.008** | 0.84 | 76.7 | 0.080 | 0.32 | 90.4 | 0.033 | 36.65M | 2,027.67 |
| Posterior Net | 0.360 | 93.1 | 0.112 | 1.06 | 75.2 | 0.139 | 0.42 | 87.9 | 0.053 | 36.60M | 1,162.48 |
| Heteroscedastic | 0.156 | 96.0 | 0.023 | 1.05 | 76.1 | 0.154 | 0.38 | 90.1 | 0.056 | 36.54M | 560.43 |
| SNGP | 0.134 | 96.0 | 0.007 | 0.74 | 78.5 | 0.078 | 0.43 | 89.7 | 0.064 | 37.50M | 916.26 |
| MIMO | 0.123 | 96.4 | 0.010 | 0.93 | 76.6 | 0.112 | 0.35 | 90.1 | 0.037 | 36.51M | 701.66 |
| BatchEnsemble | 0.136 | 96.3 | 0.018 | 0.97 | 77.8 | 0.124 | 0.35 | 90.6 | 0.048 | 36.58M | 1,498.01 |
| Deep Ensembles | **0.114** | **96.6** | 0.010 | 0.81 | 77.9 | 0.087 | 0.28 | **92.2** | 0.025 | 145.99M | 1,520.34 |
| **Density-Softmax** | 0.137 | 96.0 | 0.010 | **0.68** | **79.2** | **0.060** | **0.26** | 91.6 | **0.016** | 36.58M | 520.53 |

**Table 3:** Results for Wide Resnet-28-10 on CIFAR-100: cNLL, cAcc, and cECE are for CIFAR-100-C. AUPR-S and AUPR-C are AUPR for OOD detection on SVHN and CIFAR-10 (higher are better).

| Method | NLL(↓) | Acc(↑) | ECE(↓) | cNLL(↓) | cAcc(↑) | cECE(↓) | AUPR-S(↑) | AUPR-C(↑) | #Params(↓) | Latency(↓) |
|---|---|---|---|---|---|---|---|---|---|---|
| ERM | 0.875 | 79.8 | 0.086 | 2.70 | 51.4 | 0.239 | 0.882 | 0.745 | **36.55M** | **521.15** |
| MC Dropout | 0.785 | 80.7 | 0.049 | 2.73 | 46.2 | 0.207 | 0.832 | 0.757 | 36.55M | 4,339.03 |
| MFVI BNN | 0.944 | 77.8 | 0.097 | 3.18 | 48.2 | 0.271 | 0.882 | 0.748 | 73.07M | 818.31 |
| Rank-1 BNN | 0.692 | 81.3 | **0.018** | 2.24 | 53.8 | 0.117 | 0.884 | 0.797 | 36.71M | 2,048.90 |
| Posterior Net | 2.021 | 77.3 | 0.391 | 3.12 | 48.3 | 0.281 | 0.880 | 0.760 | 36.60M | 11,243.84 |
| Heteroscedastic | 0.833 | 80.2 | 0.059 | 2.40 | 52.1 | 0.177 | 0.881 | 0.752 | 37.00M | 568.17 |
| SNGP | 0.805 | 80.2 | 0.020 | 2.02 | 54.6 | 0.092 | **0.923** | 0.801 | 37.50M | 926.99 |
| MIMO | 0.690 | 82.0 | 0.022 | 2.28 | 53.7 | 0.129 | 0.885 | 0.760 | 36.68M | 718.11 |
| BatchEnsemble | 0.690 | 81.9 | 0.027 | 2.56 | 53.1 | 0.149 | 0.870 | 0.757 | 36.63M | 1,568.77 |
| Deep Ensembles | **0.666** | **82.7** | 0.021 | 2.27 | 54.1 | 0.138 | 0.888 | 0.780 | 146.22M | 1,569.23 |
| **Density-Softmax** | 0.780 | 80.8 | 0.038 | **1.96** | **54.7** | **0.089** | 0.910 | **0.804** | 36.64M | 522.94 |

**Table 4:** Results for Resnet-50 on ImageNet: cNLL, cAcc, and cECE are for ImageNet-C.

| Method | NLL(↓) | Acc(↑) | ECE(↓) | cNLL(↓) | cAcc(↑) | cECE(↓) | #Params(↓) | Latency(↓) |
|---|---|---|---|---|---|---|---|---|
| ERM | 0.939 | 76.2 | 0.032 | 3.21 | 40.5 | 0.103 | **25.61M** | **299.81** |
| MC Dropout | 0.919 | 76.6 | 0.026 | 2.96 | 42.4 | 0.046 | 25.61M | 601.15 |
| Rank-1 BNN | 0.886 | 77.3 | 0.017 | 2.95 | 42.9 | 0.054 | 26.35M | 990.14 |
| Heteroscedastic | 0.898 | 77.5 | 0.033 | 3.20 | 42.4 | 0.111 | 58.39M | 337.50 |
| SNGP | 0.931 | 76.1 | **0.013** | 3.03 | 41.1 | 0.045 | 26.60M | 606.11 |
| MIMO | 0.887 | 77.5 | 0.037 | 3.03 | 43.3 | 0.106 | 27.67M | 367.17 |
| BatchEnsemble | 0.922 | 76.8 | 0.037 | 3.09 | 41.9 | 0.089 | 25.82M | 696.81 |
| Deep Ensembles | **0.857** | **77.9** | 0.017 | 2.82 | **44.9** | 0.047 | 102.44M | 701.34 |
| **Density-Softmax** | 0.885 | 77.5 | 0.019 | **2.81** | 44.6 | **0.042** | 25.88M | 299.90 |

## 5.2 ROBUSTNESS PERFORMANCE

**Density-Softmax achieves competitive robust generalization with SOTA.** Tab. 2, 3, and 4 show benchmark results across shifted datasets. We observe Density-Softmax achieve a competitive result with the SOTA in NLL and accuracy under distribution shifts. Specifically, our method has the lowest NLL and highest accuracy with $0.68, 79.2\%, 1.96, 54.7\%$, and $2.81, 44.6\%$ respectively in the corrupted OOD datasets CIFAR-10-C, CIFAR-100-C, and ImageNet-C. Regarding the real-world shift CIFAR-10.1, it also achieves the lowest NLL with 0.26, and $91.6\%$ in accuracy, higher than other methods and only lower than Deep Ensembles by $0.6\%$. It is also worth noticing that our method can still preserve a high accuracy in IID at the same time and outperforms many baselines. E.g., it achieves $77.5\%$ in ImageNet, higher than Deterministc ERM, Rank-1 BNN, SNGP, BatchEnsemble, etc. More details about benchmark comparison are in Apd. C.4.2.

## 5.3 UNCERTAINTY PERFORMANCE

**Density-Softmax achieves competitive uncertainty performances with SOTA.** From the tables, we also observe our model has a competitive uncertainty quality and even sometimes outperforms

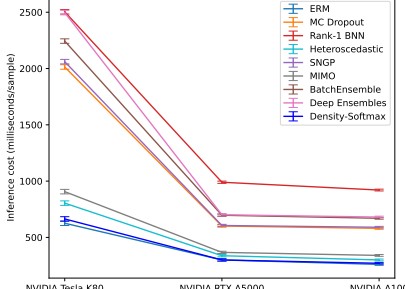

**Figure 3:** Reliability diagram of Density-Softmax v.s. different approaches, trained on CIFAR-10, test on CIFAR-10.1 (v6). **Density-Softmax is better-calibrated than others**. More Figs. are in Apd. C.4.3.

the SOTA like Rank-1 BNN, SNGP, and Deep Ensembles, especially under OOD settings. E.g., it achieves the lowest cECE with $0.060$ in CIFAR-10-C, $0.089$ in CIFAR-100-C, and $0.042$ in ImageNet-C. Similarly, it has the best AUPR-C with $0.804$ in CIFAR-10 and $0.016$ oECE in CIFAR-10.1. To take a closer look at the calibration, we visualize reliability diagrams based on the ECE in Fig. 3. We observe that Density-Softmax is better calibrated than other methods in the real-world shifted OOD test set. E.g., compared to Deterministic ERM, our model is less over-confident, *confirming Prop. 4.7*. Meanwhile, compared to Rank-1 BNN and Deep Ensembles, it is less under-confidence. Calibration details with reliability diagrams in both IID and OOD are in Apd. C.4.3.

**Density-Softmax achieves distance awareness.** From Fig. 1, we observe that our model achieves distance awareness by having uniform class probability and high uncertainty value on OOD data on the two moons dataset, *confirming Prop. 4.5*. Meanwhile, Deterministic ERM, MC Dropout, Rank-1 BNN, and Ensembles can not provide distance awareness by no informative variance for OOD data. Similar observations are in Apd. C.4.1 with different uncertainty metrics and datasets.

**Density-Softmax produces high entropy on OOD and low entropy on IID data under semantic shift**. Fig. 4 compares the density of predictive entropy between different methods trained on CIFAR-10 and tested on CIFAR-100. Because this is the semantic shift (Tran et al., 2022), we would expect the model to provide a high entropy value. Indeed, we observe that Density-Softmax achieves the highest entropy with a high-density value. In Fig. 21 in Apd. C.4.4, we also observe that our model preserves low entropy with high-density value on IID data, confirming the hypothesis that our framework can enhance uncertainty quantification. More details about the histograms and analysis are in Apd. C.4.4.

**Figure 4:** PDF plot of predictive entropy $\mathrm{H}(p(y|x))$ for the semantic shift. **Density-Softmax provide highest entropy with high density for OOD**.

## 5.4 Test-time Efficiency

**Density-Softmax outperforms SOTA in inference speed.** Our method only requires one forward pass to make a prediction, so it outperforms other SOTA in terms of inference speed. For every dataset with different backbones, we observe that Density-Softmax achieves almost the same latency with Deterministic ERM. In particular, it takes less than $525$ and $300$ ms/sample in Wide Resnet-28-10 and Resnet-50 for an inference on RTX A5000. To make a further comparison on test-time latency, we make a benchmarking across 3 modern GPU architectures in Fig. 5. We observe that our model consistency outperforms SOTA, especially for lower computational hardware like NVIDIA Tesla K80.

Having a similar latency, the tables show our model is also more reliable than Deterministic ERM by always

**Figure 5:** Inference cost comparison at test-time on ImageNet. **Density-Softmax consistently outperforms SOTA across different modern GPU architectures.**

achieving a lower NLL, ECE, and higher accuracy. *Therefore, these results suggest Density-Softmax could be a potential deterministic approach for uncertainty and robustness in real-time applications.*

**Density-Softmax outperforms SOTA in storage requirements.** From the tables, the parameters and latency of $p(Z; \alpha)$ can be measured by the minus of ours to Deterministic ERM by Re. 3.4. We observe that Density-Softmax is very lightweight with less than 36.65M parameters in Wide Resnet-

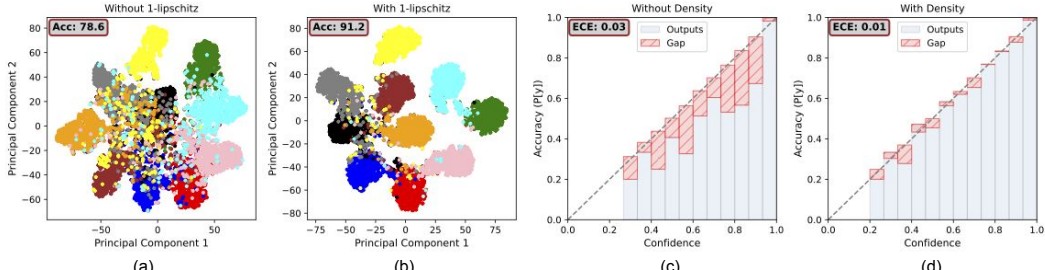

**Figure 6:** Feature visualizations comparison between models with & without 1-Lipschitz constraint on CIFAR-10-C (a & b), reliability diagrams between models with & without the density-function on CIFAR-10 (c & d).

28-10 and 25.88M in Resnet-50. These numbers are lower than other SOTA baselines, e.g., Rank-1 BNN, Heteroscedastic, SNGP, MIMO, and Deep Ensembles with Resnet-50 on ImagetNet.
As a result, combined with the latency performance, Density-Softmax outperforms other SOTA approaches in computational efficiency at the test-time. In particular, our model is *less than Deep Ensembles by 4 times in the number of parameters*. Regarding the latency, Density-Softmax is also *much faster than other SOTA baselines across different hardware architectures*.

### 5.5 ABLATION STUDY: ANALYSIS OF DENSITY SOFTMAX'S COMPONENT

**How does Density Softmax work?** Our framework is a combination of 2 main components: (1) the Lipschitz-constrained $f$ and (2) the density function $p(Z; \alpha)$. To test their importance, we make a comparison between architectures with and without each in our model. From Fig. 6, we observe that without the 1-Lipschitz regularization, the model has a worse feature representation than (1), e.g., leading to a drop in the accuracy from $91.2\%$ to $78.6\%$ on the CIFAR-10-C with defocus_blur_5. Similarly, without $p(Z; \alpha)$, the model has a worse uncertainty quality than (2), e.g., causing an increase in the ECE from 0.01 to 0.03 on CIFAR-10. These results prove that the Lipschitz-constrained $f$ helps our model improve robustness, while the density function $p(Z; \alpha)$ enhances the uncertainty quantification. Last but not least, we discover that Lipschitz-constrained $f$ not only improves the robustness but also the uncertainty quality by observing a worse ECE performance without the Lipschitz constraint, i.e., $\lambda = 0$ in Tab. 5.

**Our density model can capture distributional shifts.** Density-Softmax estimates on the low-dimensional feature space $\mathcal{Z}$, which is fixed after the first step of Algo. 1. This feature structure is low-dimensional, task-specific, and encodes meaningful semantic features (Charpentier et al., 2022), therefore much simpler to estimate when compared to complex image pixels space $\mathcal{X}$. We visualize the likelihood histogram of our Normalizing-Flows density function across training, IID testing, and OOD sets in Fig. 7. We observe that our density function provides a high likelihood for IID while low values for OOD test set. Importantly, when the shift intensity increases, the likelihood also decreases, showing our model can reduce certainty correspondingly.

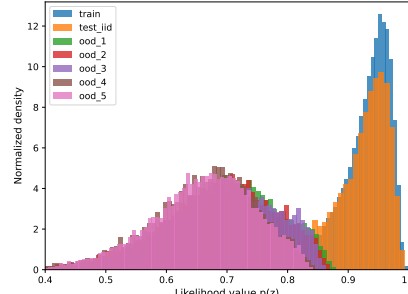

**Figure 7:** Histogram of $p(z; \alpha)$'s likelihood. Blue represents on CIFAR-10 train, Orange is IID test, Green, Red, Purple, Brown, Pink are OOD from 1-5 shift levels on CIFAR-10-C. **It produces high values on IID, lower values on OOD w.r.t. intensity levels.**

## 6 CONCLUSION

Despite showing success in reliable DNN, sampling-based methods like Deep Ensembles and BNN suffer from huge computational burdens at test-time. To tackle this challenge, we introduce Density-Softmax, a simple sampling-free approach to improve uncertainty and robustness via a combination of a feature density function from the Lipschitz-constrained feature extractor with the softmax layer. We complement this algorithm with a theoretical analysis establishing guarantees on the 1-Lipschitz constraint, solution of the minimax uncertainty risk, distance awareness on feature space, and reducing over-confidence of the standard softmax. Empirically, we find our proposed framework achieves competitive results with SOTA methods in uncertainty and robustness while outperforming them significantly in terms of memory and inference cost at test-time. We believe Density-Softmax is a balanced framework between 3 criteria: uncertainty, robustness, and test-time efficiency. We hope that our work will be an option for developers to try in real-world applications and inspire researchers to further progress in the area of improving the DNN model efficiency and reliability.

## 7    ETHICS STATEMENT

**Broader impacts.** Uncertainty and robustness are critical problems in trustworthy AI. There has been growing interest in using sampling-based methods to ensure deep learning systems are robust and reliable. Challenges often arise when deploying such models in real-world applications. In this regard, Density-Softmax significantly improves test-time efficiency while preserving reliability. This could be particularly beneficial in high-stake applications (e.g., healthcare, finance, policy decision-making, etc.), where the trained model needs to be deployed and inference on low-resource hardware or real-time response software.

**Limitations**:

1. **Density model performance in practice.** The uncertainty quality of Density-Softmax depends on the density function. Our results show if the likelihood on test OOD feature is lower than IID set, then Density-Softmax can reduce the over-confidence of the standard softmax. Yet, it can be a risk that our model might not fully capture the real-world complexity by estimating density function is not always trivial in practice (Nalisnick et al., 2019; Charpentier et al., 2022).
2. **Training cost with 1-Lipschitz regularization.** Despite showing success in test-time efficiency, we raise awareness about the challenge of training Density-Softmax on low computational infrastructure. Specifically, Density-Softmax requires a longer-time and higher-memory cost than Deterministic ERM at training-time (e.g., Fig. 8), due to 3 separate training steps and the Jacobian matrix in the regularization at the first step of Algo. 1. This 1-Lipschitz regularization also requires pre-defining the hyper-parameter $\lambda$ before training in implementation (e.g., Tab. 5).

**Remediation.** Given the aforementioned limitations, we encourage people who extend our work to: (1) proactively confront the model design and parameters to desired behaviors in real-world use cases; (2) be aware of the training challenge and prepare enough time and resources (e.g., setting enough GPU servers, training on GPU cloud services, etc.) to pre-train our framework in practice.

**Future work.** We plan to tackle Density-Softmax's limitations, including extending to regression tasks, developing new techniques to avoid computing Jacobian matrix at training-time, improving estimation techniques to enhance the quality of the density function, and continuing to reduce the number of parameters to deploy this framework in real-world systems.

## 8    REPRODUCIBILITY STATEMENT

We confirm that all our results are reproducible. Our source code is in the attached zip file as well as in the GitHub anonymous link in the main paper. Our code inherits from the uncertainty-baselines codebase, so our reported results could be referred from this page. In Apd. C, we provide detailed information about our experiments, including dataset in Apd. C.1, baseline in Apd. C.2, implementation in Apd. C.3, and additional results in Apd. C.4. The additional results in Apd. C.4 contain results for the toy dataset in Apd. C.4.1, results for the benchmark dataset in Apd. C.4.2, uncertainty details about calibration in Apd. C.4.3, and about predictive entropy in Apd. C.4.4. In Apd. B, we make further discussions about our method, including density estimation and likelihood value implementation in Apd. B.1, distance preserving and 1-Lipschitz constraint in Apd. B.2, and additional ablation study about the gradient-penalty and training cost in Apd. B.3. Finally, in Apd. A, we provide the proofs for all the results in the main paper, including proof of Thm. 4.3 in Apd. A.1, proof of Cor. 4.4 in Apd. A.2, proof of Prop. 4.5 in Apd. A.3, and proof of Prop. 4.7 in Apd. A.4.

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

## ACRONYMS

**AUPR** Area Under the Precision-Recall.
**AUROC** Area Under the Receiver Operating Characteristic.

**BNN** Bayesian Neural Network.

**DNN** Deep Neural Network.

**ECE** Expected Calibration Error.
**ERM** Empirical Risk Minimization.

**IID** Independent-identically-distributed.

**MC** Monte-Carlo.
**MFVI** Mean-Field Variational Inference.
**MIMO** Multi-input Multi-output.
**MLE** Maximum Likelihood Estimation.

**NLL** Negative log-likelihood.

**OOD** Out-of-Distribution.

**SNGP** Spectral-normalized Neural Gaussian Process.
**SOTA** State-of-the-art.

## A PROOFS

In this appendix, we provide proof of the theoretical results from the main paper.

### A.1 PROOF OF THEOREM 4.3

The proof contains two parts. The first part shows Density-Softmax provides a uniform prediction when $d(z_{ood}, Z_s) \to \infty$. The second part shows Density-Softmax preserves the in-domain prediction when $d(z_{iid}, Z_s) \to 0$.

*Part (1). Density-Softmax provides a uniform prediction when $d(z_{ood}, Z_s) \to \infty$:*

*Proof.* Consider the non-uniform in predictive distribution $\sigma(g(f(x_{ood}))) \neq \mathbb{U}$, we have

$$p(y = i|x_{ood}) = \frac{\exp(z_{ood}^\top \theta_{g_i})}{\sum_{j=1}^{K} \exp(z_{ood}^\top \theta_{g_j})} \neq \frac{1}{K}, \forall i \in \mathcal{Y}, \tag{6}$$

where $z_{ood} = f(x_{ood})$ is the latent presentation for test sample $x_{ood}$, $K$ is the total of the number of categorical in the discrete label space $\mathcal{Y}$.

Let us rewrite the Density-Softmax predictive distribution $\sigma(p(z_{ood}; \alpha) * g(z_{ood}))$ by

$$p(y = i|x_{ood}) = \frac{\exp(p(z_{ood}; \alpha) * (z_{ood}^\top \theta_{g_i})}{\sum_{j=1}^{K} \exp(p(z_{ood}; \alpha) * (z_{ood}^\top \theta_{g_j}))}, \forall i \in \mathcal{Y}. \tag{7}$$

Using Lemma 4.2, we have $\lim_{d(z_t, Z_s) \to \infty} p(z_t; \alpha) \to 0$, i.e., if $d(z_{ood}, Z_s) \to \infty$ then $p(z_{ood}; \alpha) \to 0$. Therefore, we obtain

$$\lim_{p(z_{ood}; \alpha) \to 0} \exp(p(z_{ood}; \alpha) * (z_{ood}^\top \theta_{g_i})) = e^0 = 1, \forall i \in \mathcal{Y}. \tag{8}$$

Since $\exp(p(z_{ood}; \alpha) * (z_{ood}^\top \theta_{g_j})) = 1, \forall j \in \mathcal{Y}$ when $p(z_{ood}; \alpha) \to 0$, then

$$p(y = i|x_{ood}) = \frac{\exp(p(z_{ood}; \alpha) * (z_{ood}^\top \theta_{g_i}))}{\sum_{j=1}^{K} \exp(p(z_{ood}; \alpha) * (z_{ood}^\top \theta_{g_j}))} = \frac{1}{\sum_{i=1}^{K} 1} = \frac{1}{K}, \forall i \in \mathcal{Y}. \tag{9}$$

As a consequence, when $d(z_{ood}, Z_s) \to \infty$, we obtain the conclusion: $\sigma(p(z_{ood}; \alpha) * g(z_{ood})) = \mathbb{U}$, where $\mathbb{U}$ stands for uniform distribution of Theorem 4.3.

*Part (2). Density-Softmax preserves the in-domain prediction when $d(z_{iid}, Z_s) \to 0$.*

Consider Density-Softmax predictive distribution $\sigma(p(z_{iid}; \alpha) * g(z_{iid}))$, we have

$$p(y = i|x_{iid}) = \frac{\exp(p(z_{iid}; \alpha) * (z_{iid}^\top \theta_{g_i}))}{\sum_{j=1}^{K} \exp(p(z_{iid}; \alpha) * (z_{iid}^\top \theta_{g_j}))}, \forall i \in \mathcal{Y}. \tag{10}$$

Due to the likelihood value of $p(Z; \alpha)$ is scale in the range of $(0, 1]$, we have $\lim_{d(z_t, Z_s) \to 0} p(z_t; \alpha) \to 1$, i.e., if $d(z_{iid}, Z_s) \to 0$ then $p(z_{iid}; \alpha) \to 1$. Therefore, we obtain

$$\lim_{p(z_{iid}; \alpha) \to 1} \exp(p(z_{iid}; \alpha) * (z_{iid}^\top \theta_{g_i})) = \exp(z_{iid}^\top \theta_{g_i}), \forall i \in \mathcal{Y}. \tag{11}$$

Since $\exp(p(z_{iid}; \alpha) * (z_{iid}^\top \theta_{g_i})) = \exp(z_{iid}^\top \theta_{g_i}), \forall i \in \mathcal{Y}$ when $p(z_{iid}; \alpha) \to 1$, then

$$p(y = i|x_{iid}) = \frac{\exp(p(z_{iid}; \alpha) * (z_{iid}^\top \theta_{g_i}))}{\sum_{j=1}^{K} \exp(p(z_{iid}; \alpha) * (z_{iid}^\top \theta_{g_j}))} = \frac{\exp(z_{iid}^\top \theta_{g_i})}{\sum_{j=1}^{K} \exp(z_{iid}^\top \theta_{g_j})}, \forall i \in \mathcal{Y}. \tag{12}$$

As a consequence, when $d(z_{iid}, Z_s) \to 0$, we obtain the conclusion: $\sigma(p(z_{iid}; \alpha) * g(z_{iid})) = \sigma(g(f(x_{iid})))$ of Theorem 4.3. $\qquad \square$

## A.2 PROOF OF COROLLARY 4.4

*Proof.* This proof is based on the following provable Lemma of Liu et al. (2020a):

**Lemma A.1.** *(Liu et al., 2020a; Grünwald & Dawid, 2004) (**The uniform distribution $\mathbb{U}$ is the optimal for minimax Bregman score in $x \notin \mathcal{X}_{iid}$**). Consider the Bregman score (Parry et al., 2012) as follows*

$$s(p, p^*|x) = \sum_{k=1}^{K} \left\{ [p^*(y_k|x) - p(y_k|x)] \, \psi'(p^*(y_k|x)) - \psi(p^*(y_k|x)) \right\},$$

*where $\psi$ is a strictly concave and differentiable function. Bregman score reduces to the log score when $\psi(p) = p \log(p)$, and reduces to the Brier score when $\psi(p) = p^2 - \frac{1}{K}$.*

So, let consider the strictly proper scoring rules $S(\mathbb{P}(Y|X), \mathbb{P}^*(Y|X))$ (Gneiting & Raftery, 2007; Bröcker, 2009), since $\mathbb{P}(Y|X)$ is predictive, and $\mathbb{P}^*(Y|X)$ is the data-generation distribution, then for $\mathcal{X}_{ood} = \mathcal{X}/\mathcal{X}_{idd}$, using the result from Liu et al. (2020a), we have

$$S(\mathbb{P}, \mathbb{P}^*) = \mathbb{E}_{x \sim X}(s(p, p^*|x)) = \int_{\mathcal{X}} s(p, p^*|x) p^*(x) dx \tag{13}$$

$$= \int_{\mathcal{X}} s(p, p^*|x) \left[ p^*(x|x \in \mathcal{X}_{iid}) p^*(x \in \mathcal{X}_{iid}) + p^*(x|x \in \mathcal{X}_{ood}) p^*(x \in \mathcal{X}_{ood}) \right] dx \tag{14}$$

$$= \underbrace{\mathbb{E}_{x \sim X_{iid}}(s(p, p^*|x))}_{S_{iid}(\mathbb{P}, \mathbb{P}^*)} p^*(x \in \mathcal{X}_{iid}) + \underbrace{\mathbb{E}_{x \sim X_{ood}}(s(p, p^*|x))}_{S_{ood}(\mathbb{P}, \mathbb{P}^*)} p^*(x \in \mathcal{X}_{ood}). \tag{15}$$

Therefore, since $S_{iid}(\mathbb{P}, \mathbb{P}^*)$ and $S_{ood}(\mathbb{P}, \mathbb{P}^*)$ has disjoint support, we have the minimax uncertainty risk, i.e., $\inf_{\mathbb{P}(Y|X) \in \mathcal{P}} \left[ \sup_{\mathbb{P}^*(Y|X) \in \mathcal{P}*} S(\mathbb{P}(Y|X), \mathbb{P}^*(Y|X)) \right]$ equivalents to

$$\inf_{\mathbb{P}} \sup_{\mathbb{P}^*} S(\mathbb{P}, \mathbb{P}^*) = \inf_{\mathbb{P}} \left[ \sup_{\mathbb{P}^*} [S_{iid}(\mathbb{P}, \mathbb{P}^*)] * \mathbb{P}^*(X_{iid}) + \sup_{\mathbb{P}^*} [S_{ood}(\mathbb{P}, \mathbb{P}^*)] * \mathbb{P}^*(X_{ood}) \right] \tag{16}$$

$$= \inf_{\mathbb{P}} \sup_{\mathbb{P}^*} [S_{iid}(\mathbb{P}, \mathbb{P}^*)] * \mathbb{P}^*(X_{iid}) + \inf_{\mathbb{P}} \sup_{\mathbb{P}^*} [S_{ood}(\mathbb{P}, \mathbb{P}^*)] * \mathbb{P}^*(X_{ood}). \tag{17}$$

Due to $\mathbb{P}(Y|X_{iid})$ is the predictive distribution learned from data, we obtain $\inf_{\mathbb{P}} \sup_{\mathbb{P}^*} [S_{iid}(\mathbb{P}, \mathbb{P}^*)]$ is fixed and Density-Softmax satisfy by it is the model's predictive distribution learned from IID data. On the other hand, we have

$$\sup_{\mathbb{P}^* \in \mathcal{P}^*} [S_{ood}(\mathbb{P}, \mathbb{P}^*)] = \mathbb{E}_{x \sim X_{ood}} \sup_{p^*} [s(p, p^*|x)] p(x). \tag{18}$$

Applying the result from Lemma A.1 and combining with the result for OOD data from Theorem 4.3 and Proof A.1, we obtain

$$\sigma(p(f(x_{ood}); \alpha) * g(f(x_{ood})) = \mathbb{U} = \arg \inf_{\mathbb{P} \in \mathcal{P}} \sup_{\mathbb{P}^* \in \mathcal{P}^*} [S_{ood}(\mathbb{P}, \mathbb{P}^*)]. \tag{19}$$

As a consequence, we obtain the conclusion: Density-Softmax's prediction is the optimal solution of the minimax uncertainty risk, i.e,

$$\sigma(p(f(X); \alpha) * g(f(X)) = \arg \inf_{\mathbb{P}(Y|X) \in \mathcal{P}} \left[ \sup_{\mathbb{P}^*(Y|X) \in \mathcal{P}*} S(\mathbb{P}(Y|X), \mathbb{P}^*(Y|X)) \right] \tag{20}$$

of Corollary 4.4. □

## A.3 PROOF OF PROPOSITION 4.5

*Proof.* The proof contains three parts. The first part shows density function $p(z_t; \alpha)$ is monotonically decreasing w.r.t. distance function $\mathbb{E} \|z_t - Z_s\|_{\mathcal{Z}}$. The second part shows the metric $u(x_t)$ is maximized if $p(z_t; \alpha) \to 0$. The third part shows $u(x_t)$ monotonically decreasing w.r.t. $p(z_t; \alpha)$ on the interval $(0, 1]$.

*Part (1). The monotonic decrease of density function $p(z_t; \alpha)$ w.r.t. distance function $\mathbb{E} \|z_t - Z_s\|_{\mathcal{Z}}$:*
Consider the probability density function $p(z_t; \alpha)$ follows Normalizing-Flows which output the Gaussian distribution with mean (median) $\mu$ and standard deviation $\sigma$, then we have

$$p(z_t; \alpha) = \frac{1}{\sigma\sqrt{2\pi}} \exp\left(\frac{-1}{2}\left(\frac{z_t - \mu}{\sigma}\right)^2\right). \tag{21}$$

Take derivative, we obtain

$$\frac{d}{dz_t} p(z_t; \alpha) = \left[\frac{-1}{2}\left(\frac{z_t - \mu}{\sigma}\right)^2\right]' p(z_t; \alpha) = \frac{\mu - z_t}{\sigma^2} p(z_t; \alpha) \Rightarrow \begin{cases} \frac{d}{dz_t} p(z_t; \alpha) > 0 & \text{if } z_t < \mu, \\ \\ \frac{d}{dz_t} p(z_t; \alpha) = 0 & \text{if } z_t = \mu, \\ \\ \frac{d}{dz_t} p(z_t; \alpha) < 0 & \text{if } z_t > \mu. \end{cases} \tag{22}$$

Consider the distance function $\mathbb{E} \|z_t - Z_s\|_{\mathcal{Z}}$ follows the absolute norm, then we have

$$\mathbb{E} \|z_t - Z_s\|_{\mathcal{Z}} = \mathbb{E}\left(|z_t - Z_s|\right) = \int_{-\infty}^{z_t} \mathbb{P}(Z_s \leq t) dt + \int_{z_t}^{+\infty} \mathbb{P}(Z_s \geq t) dt. \tag{23}$$

Take derivative, we obtain

$$\frac{d}{dz_t} \mathbb{E} \|z_t - Z_s\|_{\mathcal{Z}} = \mathbb{P}(Z_s \leq z_t) - \mathbb{P}(Z_s \geq z_t) \Rightarrow \begin{cases} \frac{d}{dz_t} \mathbb{E} \|z_t - Z_s\|_{\mathcal{Z}} < 0 & \text{if } z_t < \mu, \\ \\ \frac{d}{dz_t} \mathbb{E} \|z_t - Z_s\|_{\mathcal{Z}} = 0 & \text{if } z_t = \mu, \\ \\ \frac{d}{dz_t} \mathbb{E} \|z_t - Z_s\|_{\mathcal{Z}} > 0 & \text{if } z_t > \mu. \end{cases} \tag{24}$$

Combining the result in Eq. 22 and Eq. 24, we have $p(z_t; \alpha)$ is maximized when $\mathbb{E} \|z_t - Z_s\|_{\mathcal{Z}}$ is minimized at the median $\mu$, $p(z_t; \alpha)$ increase when $\mathbb{E} \|z_t - Z_s\|_{\mathcal{Z}}$ decrease and vice versa. As a consequence, we obtain $p(z_t; \alpha)$ is monotonically decreasing w.r.t. distance function $\mathbb{E} \|z_t - Z_s\|_{\mathcal{Z}}$.

*Part (2). The maximum of metric $u(x_t)$:* Consider $u(x_t) = v(d(x_t, X_s))$ in Def. 4.1, let $u(x_t)$ is the entropy of predictive distribution of Density-Softmax $\sigma(p(z = f(x); \alpha) * (g \circ f(x)))$, then we have

$$u(x_t) = \mathrm{H}(\sigma(p(z_t; \alpha) * g(z_t))) \tag{25}$$

$$= -\sum_{i=1}^{K} p(y = i | \sigma(p(z_t; \alpha) * g(z_t))) \log\left(p(y = i | \sigma(p(z_t; \alpha) * g(z_t)))\right) \tag{26}$$

$$= -\sum_{i=1}^{K} \frac{\exp(p(z_t; \alpha) * (z_t^\top \theta_{g_i}))}{\sum_{j=1}^{K} \exp(p(z_t; \alpha) * (z_t^\top \theta_{g_j}))} \log\left(\frac{\exp(p(z_t; \alpha) * (z_t^\top \theta_{g_i}))}{\sum_{j=1}^{K} \exp(p(z_t; \alpha) * (z_t^\top \theta_{g_j}))}\right). \tag{27}$$

Let $a_i = \frac{\exp(p(z_t; \alpha) * (z_t^\top \theta_{g_i}))}{\sum_{j=1}^{K} \exp(p(z_t; \alpha) * (z_t^\top \theta_{g_j}))}$, then we need to find

$$(a_1, \ldots, a_K) \text{ to maximize } -\sum_{i=1}^{K}(a_i) \log(a_i) \text{ subject to } \sum_{i=1}^{K}(a_i) - 1 = 0. \tag{28}$$

Since $-\sum_{i=1}^{K}(a_i) \log(a_i)$ is an entropy function, it strictly concave on $\vec{a}$. In addition, because the constraint is $\sum_{i=1}^{K}(a_i) - 1 = 0$, the Mangasarian-Fromovitz constraint qualification holds. So, apply the Lagrange multiplier, we have the Lagrange function

$$\mathcal{L}(a_1, \ldots, a_K, \lambda) = -\sum_{i=1}^{K}(a_i) \log(a_i) - \lambda\left(\sum_{i=1}^{K}(a_i) - 1\right). \tag{29}$$

Calculate the gradient, and we obtain

$$\nabla_{a_1,\ldots,a_K,\lambda}\mathcal{L}(a_1,\ldots,a_K,\lambda) = \left(\frac{\partial\mathcal{L}}{\partial a_1},\ldots,\frac{\partial\mathcal{L}}{\partial a_K},\frac{\partial\mathcal{L}}{\partial\lambda}\right) \tag{30}$$

$$= \left(-(\log(a_1)+\frac{1}{\ln})-\lambda,\ldots,-(\log(a_k)+\frac{1}{\ln})-\lambda,\sum_{i=1}^{K}(a_i)-1\right), \tag{31}$$

and therefore

$$\nabla_{a_1,\ldots,a_K,\lambda}\mathcal{L}(a_1,\ldots,a_K,\lambda) = 0 \Leftrightarrow \begin{cases} -(\log(a_i)+\frac{1}{\ln})-\lambda = 0, \forall i \in \mathcal{Y}, \\ \\ \sum_{i=1}^{K}(a_i)-1 = 0. \end{cases} \tag{32}$$

Consider $-(\log(a_i)+\frac{1}{\ln})-\lambda = 0, \forall i \in \mathcal{Y}$, this shows that all $a_i$ are equal (because they depend on $\lambda$ only). By using $\sum_{i=1}^{K}(a_i)-1 = 0$, we find

$$a_i = \frac{\exp(p(z_t;\alpha)*(z_t^\top\theta_{g_i}))}{\sum_{j=1}^{K}\exp(p(z_t;\alpha)*(z_t^\top\theta_{g_j}))} = \frac{1}{K}, \forall i \in \mathcal{Y}. \tag{33}$$

As a consequence, $u(x_t)$ is maximized if the predictive distribution $\sigma(p(z=f(x);\alpha)*(g\circ f(x))) = \mathbb{U}$, i.e., $p(z_t;\alpha) \to 0$ which will happen if $z_t$ is OOD data (by the result in Thm. 4.3 and Proof A.1).

*Part (3). The monotonically decrease of metric $u(x_t)$ on the interval $(0,1]$*: Consider the function

$$\mathcal{F}(p(z_t;\alpha)) = -\sum_{i=1}^{K}\frac{\exp(p(z_t;\alpha)*(z_t^\top\theta_{g_i}))}{\sum_{j=1}^{K}\exp(p(z_t;\alpha)*(z_t^\top\theta_{g_j}))}\log\left(\frac{\exp(p(z_t;\alpha)*(z_t^\top\theta_{g_i}))}{\sum_{j=1}^{K}\exp(p(z_t;\alpha)*(z_t^\top\theta_{g_j}))}\right). \tag{34}$$

Let $a = p(z_t;\alpha)$, $b_i = z_t^\top\theta_{g_i}, \forall i \in \mathcal{Y}$ then $\mathcal{F}(a) = -\sum_{i=1}^{K}\frac{e^{ab_i}}{\sum_{j=1}^{K}e^{ab_j}}\log\left(\frac{e^{ab_i}}{\sum_{j=1}^{K}e^{ab_j}}\right)$, and we need to find $\frac{d}{da}\mathcal{F}$. Take derivative, we obtain

$$\frac{d}{da}\mathcal{F} = -\sum_{i=1}^{K}\left\{\left(\frac{e^{ab_i}}{\sum_{j=1}^{K}e^{ab_j}}\right)'\log\left(\frac{e^{ab_i}}{\sum_{j=1}^{K}e^{ab_j}}\right) + \frac{e^{ab_i}}{\sum_{j=1}^{K}e^{ab_j}}\left[\log\left(\frac{e^{ab_i}}{\sum_{j=1}^{K}e^{ab_j}}\right)\right]'\right\} \tag{35}$$

$$= -\sum_{i=1}^{K}\left[\left(\frac{e^{ab_i}}{\sum_{j=1}^{K}e^{ab_j}}\right)'\log\left(\frac{e^{ab_i}}{\sum_{j=1}^{K}e^{ab_j}}\right) + \frac{e^{ab_i}}{\sum_{j=1}^{K}e^{ab_j}}\left(\frac{e^{ab_i}}{\sum_{j=1}^{K}e^{ab_j}}\right)'\frac{\sum_{j=1}^{K}e^{ab_j}}{e^{ab_i}}\right] \tag{36}$$

$$= -\sum_{i=1}^{K}\left\{\frac{b_ie^{ab_i}\sum_{j=1}^{K}e^{ab_j} - e^{ab_i}\sum_{j=1}^{K}b_je^{ab_j}}{(\sum_{j=1}^{K}e^{ab_j})^2}\left[\log\left(\frac{e^{ab_i}}{\sum_{j=1}^{K}e^{ab_j}}\right)+1\right]\right\} \tag{37}$$

$$= -\sum_{i=1}^{K}\left\{\frac{\sum_{j=1}^{K}e^{a(b_i+b_j)}(b_i-b_j)}{(\sum_{j=1}^{K}e^{ab_j})^2}\left[\log\left(\frac{e^{ab_i}}{\sum_{j=1}^{K}e^{ab_j}}\right)+1\right]\right\}. \tag{38}$$

By assuming the non-uniform in the predictive distribution $\sigma(g(f(x_{ood}))) \neq \mathbb{U}$ and $a \in (0,1]$, then

$$\frac{d}{da}\mathcal{F} = -\sum_{i=1}^{K}\left\{\frac{\sum_{j=1}^{K}e^{a(b_i+b_j)}(b_i-b_j)}{(\sum_{j=1}^{K}e^{ab_j})^2}\left[\log\left(\frac{e^{ab_i}}{\sum_{j=1}^{K}e^{ab_j}}\right)+1\right]\right\} < 0, \tag{39}$$

combining with $u(x_t)$ is maximized if $a \to 0$, we obtain $u(x_t)$ decrease monotonically on the interval $(0,1]$.

Combining the result in *Part (2)*. $u(x_t)$ is maximized if $p(z_t;\alpha) \to 0$ which will happen if $z_t$ is OOD data, and the result in *Part (3)*. $u(x_t)$ is decrease monotonically w.r.t. $p(z_t;\alpha)$ on the interval

$(0, 1]$ which will happen if $x_t$ is closer to IID data since the likelihood value $p(z_t; \alpha)$ increases, we obtain the distance awareness of $p(z = f(x); \alpha) * g$.

Combining the result in *Part (1)*. $p(z_t; \alpha)$ is monotonically decreasing w.r.t. distance function $\mathbb{E} \|z_t - Z_s\|_{\mathcal{Z}}$ and the result *distance awareness* of $p(z = f(x); \alpha) * g$, we obtain the conclusion: $\sigma(p(z = f(x); \alpha) * (g \circ f(x)))$ is distance aware on latent space $\mathcal{Z}$ of Proposition 4.5. $\qquad\square$

### A.4 PROOF OF PROPOSITION 4.7

Before making the proof, let us recall the definition of distribution calibration for the forecast:

**Definition A.2.** (Dawid, 1982; Song et al., 2019; Kuleshov et al., 2018) A forecast $h$ is said to be **distributional calibrated** if $\mathbb{P}(Y = y | h(x) = W) = w(y), \forall y \in \mathcal{Y}, W \in \Delta_y$.

Intuitively, this means the forecast $h$ is well-calibrated if its outputs truthfully quantify the predictive uncertainty. E.g., if we take all data points $x$ for which the model predicts $[h(x)]_y = 0.3$, we expect $30\%$ of them to indeed take on the label $y$. To quantify distributional calibrated in Definition A.2, an approximate estimator of the Calibration Error (Murphy, 1973) in expectation was given by Naeini et al. (2015) and is still the most commonly used measure for a multi-class model. It is referred to as the **Expected Calibration Error (ECE)** of model $h : \mathcal{X} \to \Delta_y$ is defined as

$$\mathrm{ECE}(h) := \sum_{m=1}^{M} \frac{|B_m|}{N} |\mathrm{acc}(B_m) - \mathrm{conf}(B_m)|, \tag{40}$$

where $B_m$ is the set of sample indices whose confidence falls into $\left(\frac{m-1}{M}, \frac{m}{M}\right]$ in $M$ bins, $N$ is the number of samples, bin-wise mean accuracy $\mathrm{acc}(B_m) := \frac{1}{|B_m|} \sum_{i \in B_m} \mathbb{I}(\arg\max_{y \in \mathcal{Y}}[h(x_i)]_y = y_i)$, and bin-wise mean confidence $\mathrm{conf}(B_m) := \frac{1}{|B_m|} \sum_{i \in B_m} \max_{y \in \mathcal{Y}}[h(x_i)]_y$.

*Proof.* Consider the prediction of the standard softmax $\sigma(g(f(x)))$. By definition, we have

$$\begin{aligned} \sigma : \quad & \mathbb{R}^K \to \Delta_y \\ & g(f(x)) \mapsto \sigma(g(f(x))). \end{aligned}$$

Let the logit vectors of $g(f(x))$ be $u = (u_1, \cdots, u_K) \in \mathbb{R}^K$, for an arbitrary pair of classes, i.e., $\forall i, j \in \{1, \cdots, K\}$ of the logit vector $u$, assume that $u_i < u_j$. Since the predictive distribution of Desnity-Softmax is $\sigma(p(f(x); \alpha) * g(f(x)))$, we have the corresponding logit vector is

$$p(f(x); \alpha) * u = (p(f(x); \alpha) * u_1, \cdots, p(f(x); \alpha) * u_K) \in \mathbb{R}^K. \tag{41}$$

Due to $p(f(x); \alpha) \in (0, 1]$, then we obtain the following relationship holds

$$[p(f(x); \alpha) * u]_i < [p(f(x); \alpha) * u]_j, \tag{42}$$

where $[\cdot]_i$ represents the $i$-th entry of the vector. Since the order of entries in the logit vector is unchanged between the standard softmax and Density-Softmax, we obtain

$$\arg\max_{y \in \mathcal{Y}}[\sigma(g(f(x)))]_y = \arg\max_{y \in \mathcal{Y}}[\sigma(p(f(x); \alpha) * g(f(x))]_y. \tag{43}$$

As a consequence, Density-Softmax preserves the accuracy of the standard softmax by

$$\mathrm{acc}(B_m) = \frac{1}{|B_m|} \sum_{i \in B_m} \mathbb{I}(\arg\max_{y \in \mathcal{Y}}[\sigma(g(f(x_i)))]_y = y_i) \tag{44}$$

$$= \frac{1}{|B_m|} \sum_{i \in B_m} \mathbb{I}(\arg\max_{y \in \mathcal{Y}}[\sigma(p(f(x_i); \alpha) * g(f(x_i)))]_y = y_i), \tag{45}$$

where $B_m$ is the set of sample indices whose confidence falls into $\left(\frac{m-1}{M}, \frac{m}{M}\right]$ in $M$ bins.

On the other hand, since $p(f(x); \alpha) \in (0, 1]$, we also have

$$\max_{y \in \mathcal{Y}}[\sigma(p(f(x); \alpha) * g(f(x)))]_y \le \max_{y \in \mathcal{Y}}[\sigma(g(f(x)))]_y, \tag{46}$$

and this yields

$$\frac{1}{|B_m|} \sum_{i \in B_m} \max_{y \in \mathcal{Y}}[\sigma(p(f(x_i); \alpha) * g(f(x_i)))]_y \leq \frac{1}{|B_m|} \sum_{i \in B_m} \max_{y \in \mathcal{Y}}[\sigma(g(f(x_i)))]_y. \quad (47)$$

Furthermore, by assuming the predictive distribution of the standard softmax layer $\sigma(g(f(x)))$ is over-confident, i.e., $\mathrm{acc}(B_m) \leq \mathrm{conf}(B_m), \forall B_m$, then we have

$$0 \leq \frac{1}{|B_m|} \sum_{i \in B_m} \mathbb{I}(\arg\max_{y \in \mathcal{Y}}[\sigma(g(z_i))]_y = y_i) \leq \frac{1}{|B_m|} \sum_{i \in B_m} \max_{y \in \mathcal{Y}}[\sigma(g(z_i))]_y, \forall B_m. \quad (48)$$

Combining with the result in 44, in 47 and in 47 together, for $N$ number of samples, we obtain

$$\sum_{m=1}^{M} \frac{|B_m|}{N} \underbrace{\left| \frac{1}{|B_m|} \sum_{i \in B_m} \mathbb{I}(\arg\max_{y \in \mathcal{Y}}[\sigma(p(z_i; \alpha) * g(z_i))]_y = y_i) - \frac{1}{|B_m|} \sum_{i \in B_m} \max_{y \in \mathcal{Y}}[\sigma(p(z_i; \alpha) * g(z_i))]_y \right|}_{\mathrm{ECE}(\sigma((p(f;\alpha)*g)\circ f))}$$

$$(49)$$

$$\leq \sum_{m=1}^{M} \frac{|B_m|}{N} \underbrace{\left| \frac{1}{|B_m|} \sum_{i \in B_m} \mathbb{I}(\arg\max_{y \in \mathcal{Y}}[\sigma(g(z_i))]_y = y_i) - \frac{1}{|B_m|} \sum_{i \in B_m} \max_{y \in \mathcal{Y}}[\sigma(g(z_i))]_y \right|}_{\mathrm{ECE}(\sigma(g \circ f))}, \text{ where } z_i = f(x_i).$$

$$(50)$$

As a consequence, we obtain the conclusion: $\mathrm{ECE}(\sigma((p(f; \alpha) * g) \circ f) \leq \mathrm{ECE}(\sigma(g \circ f))$ of Proposition 4.7. □

## B FURTHER DISCUSSION AND ABLATION STUDY

This appendix will make a detailed discussion about our implementation to control the range of the likelihood value in Algorithm 1, the relationship between distance preserving and 1-Lipschitz constraint to improve uncertainty quality, the additional ablation study about training cost and the gradient-penalty coefficient, and the additional comparison with other sampling-free methods.

### B.1 DENSITY ESTIMATION AND LIKELIHOOD VALUE

Recall that the output in the inference step of Algorithm 1 has the form

$$p(y = i|x_t) = \frac{\exp(p(z_t; \alpha) * (z_t^\top \theta_{g_i}))}{\sum_{j=1}^{K} \exp(p(z_t; \alpha) * (z_t^\top \theta_{g_j}))}, \forall i \in \mathcal{Y}. \quad (51)$$

Let's consider: $\exp(p(z_t; \alpha) * (z_t^\top \theta_{g_i}))$, if the likelihood value of $p(z_t; \alpha)$ is too large, then the output of the exponential function may go to a very large value, leading to the computer can not store to compute. Therefore, we need to find a scaling technique to avoid this issue.

Recall that for density estimation, we use Normalizing-Flows (Dinh et al., 2017; Papamakarios et al., 2021). Instead of returning likelihood value $p(Z; \alpha)$, this estimation returns the logarithm of likelihood $\log(p(Z; \alpha))$, then we need to take exponentially to get the likelihood value by

$$p(Z; \alpha) = e^{\log(p(Z;\alpha))}. \quad (52)$$

By doing so, there will be two properties for the range of likelihood value $p(Z; \alpha)$. First, value of $p(Z; \alpha) \neq 0$ by $\log(0)$ being undefined. Second, $p(Z; \alpha)$ is always positive by the output of the exponential function is always positive. As a result, we obtain the likelihood value $p(Z; \alpha)$ is in the range of $(0, +\infty]$.

To avoid the numerical issue when $p(Z; \alpha) \to +\infty$, we use the scaling technique to scale the range $(0, +\infty]$ to $(0, 1]$ by the following formula

$$p(Z; \alpha) = \frac{p(Z; \alpha)}{\max(p(Z_{train}; \alpha))}. \quad (53)$$

---

**Algorithm 2** Scaling likelihood

---

**Scaling Input:** Training data $D_s$, encoder $f$, trained density function $p(Z; \alpha)$.
max_trainNLL $\leftarrow 0$
**for** $e = 1 \rightarrow$ epochs **do**
    Sample $D_B$ with a mini-batch $B$ for source data $D_s$
    $Z = f(X \in D_B)$
    batch_trainNLL $\leftarrow p(Z; \alpha)$
    **if** $\max($batch_trainNLL$) > $ max_trainNLL **then**
        max_trainNLL $\leftarrow \max($batch_trainNLL$)$
    **end if**
**end for**
**Scaling Inference Input:** Test sample $x_t$
$z_t = f(x_t)$
$p(z_t; \alpha) = \frac{p(z_t; \alpha)}{\text{max\_trainNLL}}$

---

It is also worth noticing that there also can be a case $e^{\log(p(Z;\alpha))}$ in Equation 52 goes to $+\infty$ if $\log(p(Z; \alpha))$ is too large, therefore, we also need to scale it to avoid the numerical issue. The pseudo-code of our scaling algorithm and inference process is presented in Algorithm 2.

## B.2 DISTANCE PRESERVING AND 1-LIPSCHITZ CONSTRAINT

To improve the uncertainty quality, Liu et al. (2020a) introduce distance awareness definition on sample space $\mathcal{X}$, which is stronger than our definition on feature space $\mathcal{Z}$. However, in order to make the model distance aware on $\mathcal{X}$, one necessary condition is $f(x)$ must be an isometric mapping, i.e., distance preserving on latent space $\mathcal{Z}$ by satisfy the bi-Lipschitz condition (Searcód, 2006)

$$L_1 * \|x_1 - x_2\|_{\mathcal{X}} \leq \|f(x_1) - f(x_2)\|_{\mathcal{Z}} \leq L_2 * \|x_1 - x_2\|_{\mathcal{X}}, \tag{54}$$

for positive and bounded constants $0 < L1 < 1 < L2$. Although our model only guarantees distance awareness on $\mathcal{Z}$ and does not hold on $\mathcal{X}$, it still aims to achieve the upper bound of Eq. 54 by the 1-Lipschitz constraint, i.e., $\|f(x_1) - f(x_2)\|_2 \leq \|x_1 - x_2\|_2$. This helps $f(x)$ assure that if the sample is similar, the feature will be similar as well, providing meaningful correspondence with the semantic properties of the input data $\mathcal{X}$. Therefore, the 1-Lipschitz not only helps to improve robustness but may also support to improve uncertainty performance. The empirical evidence is confirmed in Table 5 by the calibrated uncertainty error without 1-Lipschitz constraint ($\lambda = 0$) is significantly higher than with 1-Lipschitz constraint (e.g., $\lambda = 1e - 4$).

## B.3 ADDITIONAL ABLATION STUDY: GRADIENT-PENALTY AND TRAINING COST

| $\lambda$ | NLL($\downarrow$) | Acc($\uparrow$) | ECE($\downarrow$) | cNLL($\downarrow$) | cAcc($\uparrow$) | cECE($\downarrow$) |
|---|---|---|---|---|---|---|
| 0 | 0.142 | 96.1 | 0.015 | 0.79 | 77.0 | 0.086 |
| 1 (w/o Eq. 4) | 0.165 | 94.8 | 0.017 | 0.66 | 79.9 | 0.041 |
| 1 | 0.162 | 95.0 | 0.015 | 0.64 | 80.0 | 0.039 |
| 1e-1 | 0.159 | 95.2 | 0.014 | 0.66 | 79.3 | 0.040 |
| 1e-2 | 0.146 | 95.6 | 0.009 | 0.66 | 79.7 | 0.047 |
| 1e-3 | 0.144 | 95.7 | 0.012 | 0.66 | 79.9 | 0.051 |
| 1e-4 | 0.137 | 96.0 | 0.010 | 0.68 | 79.2 | 0.060 |

**Table 5:** Performance of Density-Softmax with different gradient-penalty coefficient $\lambda$, model is trained on CIFAR-10, tested on IID CIFAR-10 and OOD CIFAR-10-C.

Tab. 5 shows the performance of Density-Softmax across different values of the gradient-penalty hyper-parameter $\lambda$. Firstly, we observe there is a trade-off between the performance on IID and OOD data w.r.t. $\lambda$. Secondly, the fine-tuning classifier step in Eq. 4 is essential by a better performance of $\lambda = 1$ than $\lambda = 1$ (w/o Eq. 4). Finally, with an appropriate $\lambda = 1e - 4$, Density-Softmax can balance the performance between IID and OOD data. Notably, this gradient-penalty not only helps to improve the accuracy but also the uncertainty by a lower ECE than $\lambda = 0$, confirming the hypothesis that the 1-Lipschitz constraint supports improving uncertainty quality at the same time.

The gradient-penalty regularization and three separate training steps in Algorithm 1, however, make Density-Softmax have a higher cost than Deterministic ERM at training-time. That said, our training

cost is still lower than other SOTA techniques, e.g., Deep Ensembles, BatchEnsemble, MIMO, Rank-1 BNN in Fig. 8 (left). Importantly, the efficient benefits of Density-Softmax are illustrated with a much lower inference cost than other SOTA methods at test-time in Fig. 8 (right).

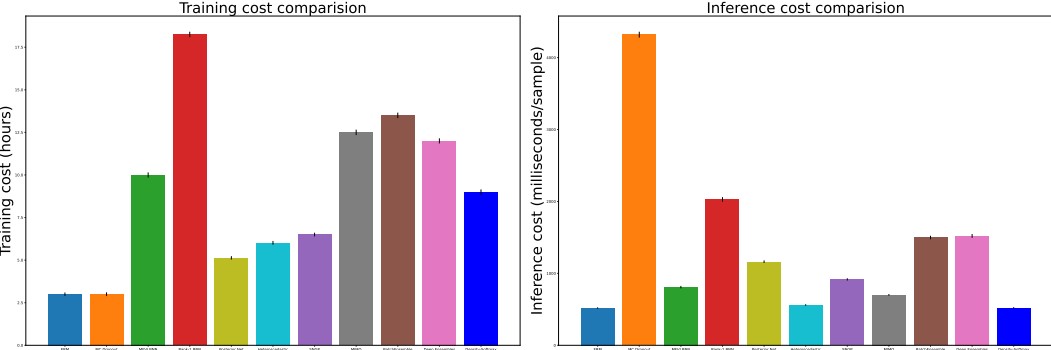

**Figure 8:** Comparison in training cost (hours) on NVIDIA A100 and inference cost (milliseconds/sample) on NVIDIA RTX A5000, with error bars across 10 seeds of Wide Resnet-28-10 on CIFAR-10 dataset. **Density-Softmax outperforms SOTA in inference speed and still has a lower training cost than some baselines.**

### B.4 ADDITIONAL COMPARISON WITH SAMPLING-FREE METHODS

**Table 6:** Additional results from Tab. 2, with DUQ (Van Amersfoort et al., 2020), DDU (without Temperature Scaling) (Mukhoti et al., 2023), DUE (van Amersfoort et al., 2022), and NatPN (Charpentier et al., 2022).

| Method | NLL(↓) | Acc(↑) | ECE(↓) | cNLL(↓) | cAcc(↑) | cECE(↓) | oNLL(↓) | oAcc(↑) | oECE(↓) | #Params(↓) | Latency(↓) |
|---|---|---|---|---|---|---|---|---|---|---|---|
| DUQ | 0.239 | 94.7 | 0.034 | 1.35 | 71.6 | 0.183 | 0.49 | 87.9 | 0.068 | 40.61M | 1,538.35 |
| DDU (w/o TS) | 0.159 | 96.0 | 0.024 | 1.06 | 76.0 | 0.153 | 0.39 | 89.8 | 0.063 | 37.60M | 954.31 |
| DUE | 0.145 | 95.6 | 0.007 | 0.84 | 77.8 | 0.079 | 0.46 | 89.2 | 0.066 | 37.50M | 916.26 |
| NatPN | 0.242 | 92.8 | 0.041 | 0.89 | 73.9 | 0.121 | 0.46 | 86.3 | 0.049 | 36.58M | 601.35 |
| Deep Ensembles | **0.114** | **96.6** | 0.010 | 0.81 | 77.9 | 0.087 | 0.28 | **92.2** | 0.025 | 145.99M | 1,520.34 |
| **Density-Softmax** | 0.137 | 96.0 | 0.010 | **0.68** | **79.2** | **0.060** | **0.26** | 91.6 | **0.016** | 36.58M | **520.53** |

**Table 7:** Additional results from Tab. 3, with DUQ (Van Amersfoort et al., 2020), DDU (without Temperature Scaling) (Mukhoti et al., 2023), DUE (van Amersfoort et al., 2022), and NatPN (Charpentier et al., 2022).

| Method | NLL(↓) | Acc(↑) | ECE(↓) | cNLL(↓) | cAcc(↑) | cECE(↓) | AUPR-S(↑) | AUPR-C(↑) | #Params(↓) | Latency(↓) |
|---|---|---|---|---|---|---|---|---|---|---|
| DUQ | 0.980 | 78.5 | 0.119 | 2.84 | 50.4 | 0.281 | 0.878 | 0.732 | 77.58M | 1,547.35 |
| DDU (w/o TS) | 0.877 | 79.7 | 0.086 | 2.70 | 51.3 | 0.240 | 0.890 | 0.797 | 37.60M | 959.25 |
| DUE | 0.902 | 79.1 | 0.038 | 2.32 | 53.5 | 0.127 | 0.897 | 0.787 | 37.50M | 926.99 |
| NatPN | 1.249 | 76.9 | 0.091 | 2.97 | 48.0 | 0.265 | 0.875 | 0.768 | 36.64M | 613.44 |
| Deep Ensembles | **0.666** | **82.7** | 0.021 | 2.27 | 54.1 | 0.138 | 0.888 | 0.780 | 146.22M | 1,569.23 |
| **Density-Softmax** | 0.780 | 80.8 | 0.038 | **1.96** | **54.7** | **0.089** | 0.910 | **0.804** | 36.64M | **522.94** |

**More detailed about comparison with NatPN in terms of the approach and algorithm.**

1. Similarities: both of us have the same idea in terms of (1) Estimating a marginal density model on feature space. (2) Combining this density model with the classifier to improve the model's uncertainty performance. (3) From the Bayesian view, both of us can be considered as a point estimate approach.

2. Differences: (1) From the Bayesian view, optimizing with NatPN is equivalent to doing Maximum a posteriori estimation (MAP) while optimizing with Density-Softmax is equivalent to doing Maximum likelihood estimation (MLE). (2) The predictive distribution in NatPN is not a standard softmax output because it is not normalized by a natural exponent function. Meanwhile, Density-Softmax is normalized by a natural exponent function.

3. Benefits of Density-Softmax versus NatPN. (1) Density-Softmax can calculate the predictive distribution without the need to consider how to define prior parameters. The prior can hurt the posterior performance in NatPN if we pre-define a bad prior. (2) The natural exponential helps Density-Softmax easily optimize with the cross-entropy loss by the nice derivative property of $\ln(\exp(a)) = a$ (Goodfellow et al., 2016), and produce a sharper distribution with a higher probability of the largest entry in the logit vector and lower probabilities of the smaller entries when compared with the normalization of NatPN.

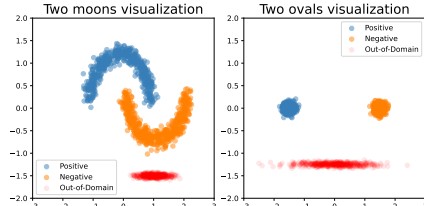

**Figure 9:** Data visualization on the Toy dataset with two moons (Left) and two ovals (Right). Training data for positive (Orange) and negative classes (Blue). OOD data (Red) are not observed during training.

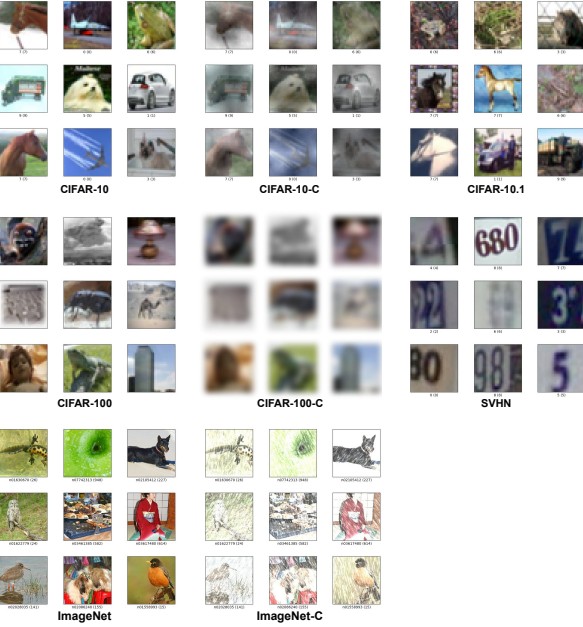

**Figure 10:** The benchmark dataset summarizations. For each dataset, we pick random images and show corresponding shifted images from a random distributional shift set. For CIFAR-10-C is fog_5, CIFAR-100-C is blurring, and ImageNet-C is snow_5. CIFAR-10.1 is the real-world distributional shift.

## C ADDITIONAL EXPERIMENTS

### C.1 DATASET DETAILS

This appendix provides more detail about the datasets used in our experiments. Figure 10 visualizes examples for the vision and language benchmark datasets. There are a total of 6 datasets (containing the toy dataset in Figure 9) widely used for classification tasks in distributional shifts, including:

- **Toy dataset (Liu et al., 2020a)** includes two types of 2D classification datasets-two moons and two ovals. For the training dataset which is represented by the two in-domain classes (500 orange and 500 blue data points), the two moons consist of two banana-shaped distributions separable by nonlinear decision boundary. In contrast, the two ovals consist of two near-flat Gaussian distributions that are separable by a linear decision boundary. For the OOD set, we generate and color by 500 red data points.

- **CIFAR-10-C (Hendrycks & Dietterich, 2019)** has $10,000$ colored samples of $\dim = (3, 32, 32)$ with 10 classes in classification problem. The distributional shifts are generated from the CIFAR-10 (Krizhevsky & Hinton, 2009) test images over a total of 15 corruptions, with 15 noise type $d \in \{$identity, gaussian noise, shot noise, impulse noise, defocus blur, frosted glass blur, motion blur, zoom blur, snow, frost, fog, brightness, contrast, elastic, pixelate, jpeg compression$\}$. For each noise, there are 5 skew intensities represented for the level noise to evaluate the robustness, so we have a total of 75 OOD test sets.

- **CIFAR-10.1 (Recht et al., 2018)** contains $2,021$ images for version v4 and $2,000$ images with class balanced for version v6, of dimension $(3, 32, 32)$ in classification problem with the same 10 classes in CIFAR-10. The dataset is collated in real-world images on the Inter-

net and is a subset of the Tiny Images (Torralba et al., 2008). This is used to additionally test the reliability of models under real-world distributional shifts.

- **CIFAR-100-C (Hendrycks & Dietterich, 2019)** has 100 classes containing 100 testing images of dimension $(3, 32, 32)$ per each class. The distributional shifts are generated from the CIFAR-100 (Krizhevsky & Hinton, 2009) test set over a total of 17 corruptions, with 17 noise type $d \in \{$identity, gaussian noise, shot noise, impulse noise, defocus blur, gaussian blur, motion blur, zoom blur, frost, fog, brightness, contrast, elastic transform, pixelate, jpeg compression, saturate, spatter, speckle noise$\}$. Similar to CIFAR-10-C, there are 5 skew intensities for each noise corruption, as a result, we have a total of 85 OOD test sets.

- **SVHN (Netzer et al., 2011)** contains 600,000 digit images coming from real-world data of dimension $(3, 32, 32)$. We use this as a far domain to test OOD detection with CIFAR.

- **ImageNet-C (Hendrycks & Dietterich, 2019)** includes $50,000$ photos of dimension $(3, 224, 224)$ in classification problem with 1000 object categories, generated from the test images of ImageNet (Deng et al., 2009) with corruptions and skew intensities similar to CIFAR-10-C. Therefore, we also have a total of 75 OOD test sets.

## C.2 BASELINE DETAILS

This appendix provides an exhaustive literature review of 10 SOTA related methods which are used to make comparisons with our model by using the uncertainty-baselines (Nado et al., 2021) (*We exclude Mixup (Zhang et al., 2018; Carratino et al., 2022) and Augmix (Hendrycks* et al., 2020) by we do not use data-augmentation*):

- **Deterministic ERM (Vapnik, 1998)** is a standard deterministic model. In test-time, it predicts the label immediately by a single forward pass.

- **MC Dropout (Gal & Ghahramani, 2016b)** includes dropout regularization method in the model. In test-time, it uses MC sampling by dropout to make different predictions, then get the mean of the list prediction.

- **MFVI BNN (Wen et al., 2018)** uses the BNN by putting distribution over the weight by mean and variance per each weight. Because each weight consists of mean and variance, the total model weights will double as the Deterministic ERM model.

- **Rank-1 BNN (Dusenberry et al., 2020)** uses the mixture approximate posterior to benefits from multimodal representation with local uncertainty around each mode, compute posterior on rank-1 matrix while keeping weight is deterministic. In test-time, it samples weight distribution by MC to make different predictions, then gets the mean of this list.

- **Posterior Net (Charpentier et al., 2020)** is based on Evidential deep learning (Sensoy et al., 2018) with a latent density function by utilizing conditional densities per class, intuitively acting as class conditionals on the latent space. Due to belonging to the Bayesian perspective, it needs to select a "good" Prior distribution, which is often difficult in practice.

- **Heteroscedastic (Collier et al., 2021)** is a probabilistic approach to modeling input-dependent by placing a multivariate Normal distributed latent variable on the final hidden layer of a neural network classifier. In test-time, it uses MC sampling to make different predictions, then get the mean of the list prediction.

- **SNGP (Liu et al., 2020a)** is a combination of the last Gaussian Process layer with Spectral Normalization to the hidden layers. Because using the Spectral Normalization for every weight in each residual layer with the power iteration method (Gouk et al., 2021; Miyato et al., 2018), and Gaussian Process layer with MC sampling, the weight and latency of SNGP is considerably higher than the Deterministic ERM model.

- **MIMO (Havasi et al., 2021)**, i.e., multi-input multi-output, it trains multiple independent subnetworks within a network. In test-time, it performs multiple independent predictions in a single forward pass.

- **BatchEnsemble (Wen et al., 2020)** defines each weight matrix to be the Hadamard product of a shared weight among ensemble members and a rank-1 matrix per member. In test-time, the final prediction is calculated from the mean of the list prediction of the ensemble.

- **Deep Ensembles (Lakshminarayanan et al., 2017)** includes multiple Deterministic ERM trained with different seeds. In test-time, the final prediction is calculated from the mean of the list prediction of the ensemble. Due to aggregates from multiple deterministic models, the total of model weights needed to store will increase linearly w.r.t. the number of models.

### C.3 IMPLEMENTATION DETAILS

In this appendix, we describe the data-processing techniques, neural network architectures, hyper-parameters, and details for reproducing our experiments. Based on the code of Nado et al. (2021), we use similar settings with their Deterministic ERM for everything for a fair comparison on the benchmark dataset. For the setting on the toy dataset, we follow the settings of Liu et al. (2020a).

**Data processing techniques.** We only use normalization to process images for the benchmark datasets. Specifically, for CIFAR-10 and CIFAR-100, we normalize by the mean is $[0.4914, 0.4822, 0.4465]$ and standard deviation is $[0.2470, 0.2435, 0.2616]$. For ImageNet, we normalize by the mean is $[0.485, 0.456, 0.406]$ and standard deviation is $[0.229, 0.224, 0.225]$ (note that we do not perform augmentation techniques of Hendrycks* et al. (2020) and Zhang et al. (2018)).

**Architectures and hyper-parameters.** We list the detailed value of hyper-parameters used for each dataset in Table 8 and the architecture of Normalizing-Flows in Table 9. For a fair comparison with deterministic, we always use the exact same hyper-parameters as the Deterministic ERM.

**Demo notebook code for Algorithm 1**

```python
import tensorflow as tf

#Define a features extractor f.
encoder = tf.keras.Sequential([
        tf.keras.layers.Dense(100, activation = "relu"),
        tf.keras.layers.Dense(100, activation = "relu"),
])
#Define a classifier g.
classifier = tf.keras.layers.Dense(10)

#Define a tf step function to pre-train model w.r.t. Eq. 2.
@tf.function
def pre_train_step(x, y):
    with tf.GradientTape(persistent = True) as tape:
        tape.watch(x)
        features = encoder(x)
        logits = classifier(features)
        loss_value = tf.cast(loss_fn(y, logits), tf.float64)
        grad_norm = tf.sqrt(tf.reduce_sum(tape.batch_jacobian(features, x)**2, axis = [1, 2]))
        loss_value += (grad_lambda * tf.reduce_mean(((grad_norm - 1)**2)))

    list_weights = encoder.trainable_weights + classifier.trainable_weights
    grads = tape.gradient(loss_value, list_weights)
    optimizer.apply_gradients(zip(grads, list_weights))
    return loss_value

#Define a tf step function to re-update the classifier by feature density model w.r.t. Eq. 4.
@tf.function
def train_step(x, y, flow_model, train_likelihood_max):
    with tf.GradientTape() as tape:
        features = encoder(x)
        likelihood = tf.exp(flow_model.score_samples(features))
        likelihood = (tf.expand_dims(likelihood, 1))/(train_likelihood_max)
        logits = classifier(features) * tf.cast(likelihood, dtype=tf.float32)
        loss_value = loss_fn(y, logits)

    grads = tape.gradient(loss_value, classifier.trainable_weights)
    optimizer.apply_gradients(zip(grads, classifier.trainable_weights))
    return loss_value

#Define a tf step function to make inference w.r.t. Eq. 5.
@tf.function
def test_step(x, flow_model, train_likelihood_max):
    features = encoder(x)
    likelihood = tf.exp(flow_model.score_samples(features))
    likelihood = (tf.expand_dims(likelihood, 1))/(train_likelihood_max)
    logits = classifier(features) * tf.cast(likelihood, dtype=tf.float32)
    y_pred = tf.nn.softmax(logits)
    return y_pred
```

**Table 8:** Condition dataset, hyper-parameters, and their default values in our experiments. Settings are inherited from Liu et al. (2020a) for the toy dataset and from Nado et al. (2021) for CIFAR-10-100 and ImageNet. **Note that we always use the exact same hyper-parameters as the Deterministic ERM for a fair comparison**.

| Conditions | Hyper-parameters | Default value |
|---|---|---|
| Toy dataset | backbone | ResFFN-12-128 (Liu et al., 2020a) |
| | epochs | 100 |
| | batch size | 128 |
| | optimizer | Adam (Kingma & Ba, 2015) |
| | learning rate | 1e-4 |
| | density estimation epochs (Flows) | 300 |
| | density optimizer (Flows) | Adam (Kingma & Ba, 2015) |
| | density learning rate (Flows) | 1e-4 |
| | re-optimize classifier epochs | 1 |
| | gradient-penalty coefficient | 1e-2 |
| CIFAR-10-100 | backbone | Wide Resnet-28-10 (Zagoruyko & Komodakis, 2016) |
| | epochs | 200 |
| | checkpoint interval | 25 |
| | batch size | 64 |
| | optimizer | SGD(momentum = 0.9, nesterov = True) |
| | learning rate | 0.05 |
| | lr decay epochs | [60, 120, 160] |
| | lr decay ratio | 0.2 |
| | L2 regularization coefficient | 2e-4 |
| | scale range | -[0.55-0.4, 0.4-0.3] |
| | density estimation epochs | 50 |
| | density optimizer | Adam (Kingma & Ba, 2015) |
| | density learning rate | 1e-4 |
| | re-optimize classifier epochs | 10 |
| | gradient-penalty coefficient | 1e-4, 1e-5 (CIFAR-10, CIFAR-100) |
| ImageNet | backbone | Resnet-50 (He et al., 2016) |
| | epochs | 90 |
| | checkpoint interval | 25 |
| | batch size | 64 |
| | optimizer | SGD(momentum = 0.9, nesterov = True) |
| | learning rate | 0.05 |
| | lr decay epochs | [30, 60, 80] |
| | lr decay ratio | 0.1 |
| | L2 regularization coefficient | 1e-4 |
| | scale range | -[0.4, 0.3] |
| | density estimation epochs | 10 |
| | density optimizer | Adam (Kingma & Ba, 2015) |
| | density learning rate | 1e-4 |
| | re-optimize classifier epochs | 2 |
| | gradient-penalty coefficient | 1e-1 |

**Table 9:** Two Coupling structures-(a) and (b) for Normalizing-Flows (Dinh et al., 2017; Papamakarios et al., 2021) on the Toy, CIFAR-10, CIFAR-100, and ImageNet. Input: $Z \in \mathbb{R}^{d_z}$, where $d_z = 128$ for ResFFN-12-128, $d_z = 640$ for Wide Resnet-28-10, and $d_z = 2048$ for Resnet-50.

| (a) $S_{\theta_S}$ | (b) $T_{\theta_T}$ |
|---|---|
| Input: $Z \in \mathbb{R}^{d_z}$ | Input: $Z \in \mathbb{R}^{d_z}$ |
| Dense (units = 16, regularizers l2 = 0.01, ReLU) | Dense (units = 16, regularizers l2 = 0.01, ReLU) |
| Dense (units = 16, regularizers l2 = 0.01, ReLU) | Dense (units = 16, regularizers l2 = 0.01, ReLU) |
| Dense (units = 16, regularizers l2 = 0.01, ReLU) | Dense (units = 16, regularizers l2 = 0.01, ReLU) |
| Dense (units = 16, regularizers l2 = 0.01, ReLU) | Dense (units = 16, regularizers l2 = 0.01, ReLU) |
| Dense (units = 16, regularizers l2 = 0.01, Tanh) | Dense (units = 16, regularizers l2 = 0.01, Linear) |

**Dataset, source code, and computing system.** The source code is provided in the mentioned GitHub in the main paper, including our notebook demo on the toy dataset, scripts to download the benchmark dataset, setup for environment configuration, and our provided code (detail in README.md). We train our model on two single GPUs: NVIDIA A100-PCIE-40GB with 8 CPUs: Intel(R) Xeon(R) Gold 6248R CPU @ 3.00GHz with 8GB RAM per each, and require 100GB available disk space for storage. We test our model on three different settings, including (1) a single GPU: NVIDIA Tesla K80 accelerator-12GB GDDR5 VRAM with 8-CPUs: Intel(R) Xeon(R) Gold 6248R CPU @ 3.00GHz with 8GB RAM per each; (2) a single GPU: NVIDIA RTX A5000-24564MiB with 8-CPUs: AMD Ryzen Threadripper 3960X 24-Core with 8GB RAM per each; and (3) a single GPU: NVIDIA A100-PCIE-40GB with 8 CPUs: Intel(R) Xeon(R) Gold 6248R CPU @ 3.00GHz with 8GB RAM per each.

### C.4 EMPIRICAL RESULT DETAILS

In this appendix, we provide additional results from the experiments, including distance-aware visualizations on the toy dataset, detailed uncertainty & robustness benchmarking between methods, calibration uncertainty, and predictive entropy on the benchmark datasets.

#### C.4.1 DISTANCE AWARENESS ON THE TOY DATASET

In the main paper, we have shown that Density-Softmax is distance-aware on two moons dataset with the variance of Bernoulli distribution surface. To test whether this observation is consistent across different settings, we continue to do experiments with different uncertainty metrics and datasets.

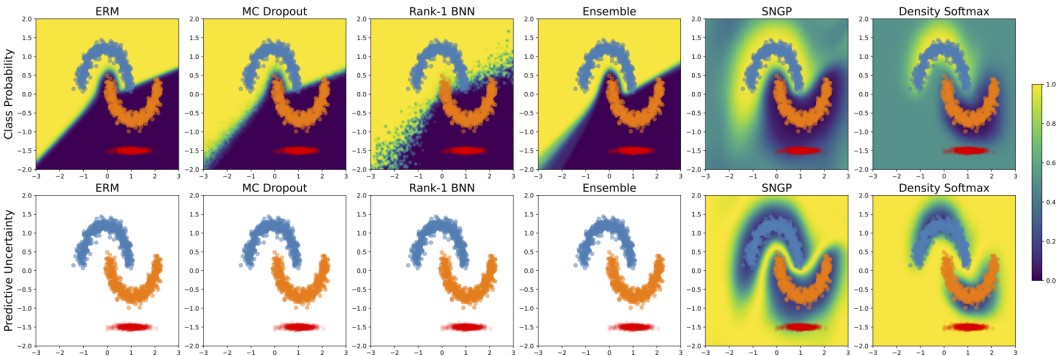

**Figure 11:** The class probability $p(y|x)$ (Top Row) and predictive entropy $H(Y|X = x) = -p(y|x) * \log_2(p(y|x)) - (1 - p(y|x)) * \log_2((1 - p(y|x)))$ surface (Bottom Row) of our Density-Softmax versus different approaches. **Our Density-Softmax achieves distance awareness with a uniform class probability and high entropy value on OOD data**. *Note that the white background due to* $\log(0)$ *is undefined.*

**Different uncertainty metrics.** Figure 11 presents a comparison in terms of predictive entropy which has the following formula

$$\text{H}(Y|X = x) = -p(y|x) * \log_2(p(y|x)) - (1 - p(y|x)) * \log_2((1 - p(y|x))). \tag{55}$$

Because of the over-confidence of Deterministic ERM, MC Dropout, Rank-1 BNN, and Ensemble which return $p(y|x) = 0$, theirs predictive entropy become undefined by $\log_2(p(y|x)) = \log_2(0)$ is undefined. As a result, we have a white background for these mentioned baselines, showing these methods are not able to be aware of the distance OOD dataset. For MC Dropout and Ensemble, since these two methods do not provide a convenient expression of the predictive variance of Bernoulli distribution, we also follow the work of Liu et al. (2020a) to plot their predictive uncertainty as the distance of the maximum predictive probability from 0.5, i.e.,

$$u(x) = 1 - 2 * |p(y|x) - 0.5|, \tag{56}$$

so that $u(x) \in [0, 1]$ in Figure 12. Although with different uncertainty metrics, Deterministic ERM, MC Dropout, Rank-1 BNN, and Ensemble still perform badly, showing truly unable to have the distance-aware ability. Meanwhile, compared with SNGP, our Density-Softmax performance is

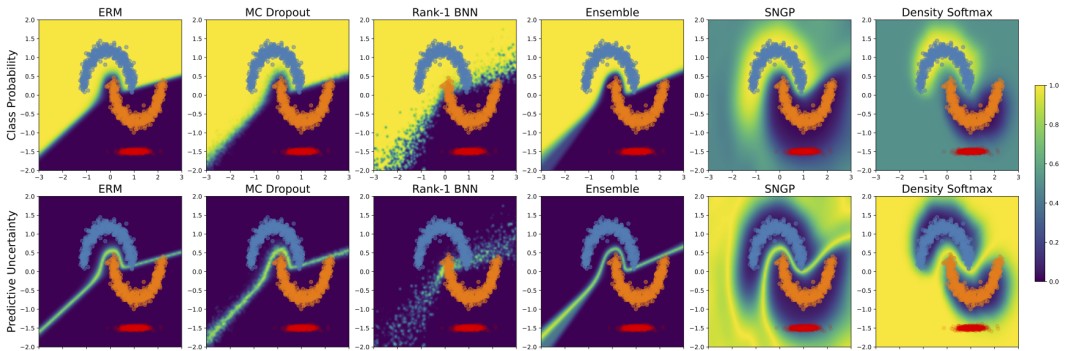

**Figure 12:** The class probability $p(y|x)$ (Top Row) and predictive uncertainty $u(x) = 1 - 2 * |p(y|x) - 0.5|$ surface (Bottom Row) of our Density-Softmax versus different approaches. **Our Density-Softmax achieves distance awareness with a uniform class probability and high uncertainty value on OOD data.**

still better with 1.0 values on both uncertainty surfaces in these two figures while SNGP still make significant mistakes with low uncertainty value on OOD data.

**Different toy dataset.** To continue to verify a consistency observation across different empirical settings, we illustrate distance-aware ability by predictive variance surface on the second toy dataset. Figure 13 compares the methods as mentioned earlier on two ovals datasets. Similar to previous settings, only SNGP is able to show its distance-aware uncertainty. However, for around half of OOD data points, its predictive uncertainty is still 0.0, showing still has limitations in distance-aware ability. In contrast, our Density-Softmax performs well, with 0.0 uncertainty value on IID training data points while always returning 1.0 on OOD data points, continue confirming the hypothesis that our Density-Softmax is a better distance-aware method.

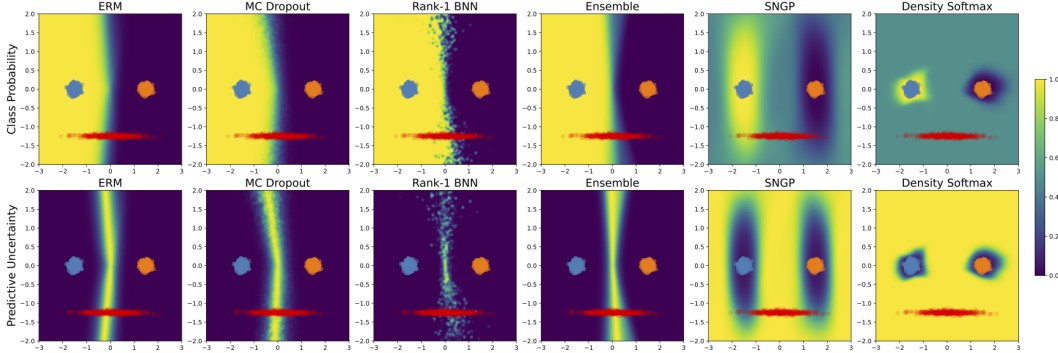

**Figure 13:** The class probability $p(y|x)$ (Top Row) and predictive uncertainty $u(x) = 1 - 2 * |p(y|x) - 0.5|$ surface (Bottom Row) of our Density-Softmax versus different approaches on the two ovals 2D classification benchmarks. **Our Density-Softmax achieves distance awareness with a uniform class probability and high uncertainty value on OOD data.**

### C.4.2 BENCHMARK RESULT DETAILS

In order to compare the uncertainty and robustness between methods in more detail, we visualize Figure 14, the box plots of the NLL, accuracy, and ECE across corruptions under distributional shifts level in CIFAR-100 based dataset from Table 3. Overall, besides achieving the highest accuracy, lowest NLL and ECE on average, our Density-Softmax also always achieves the best accuracy, NLL, and ECE performance across different shift intensities in Figure 14. For each intensity, our results also show a low variance, implying the stability of our algorithm across different corruption types. *More importantly, compared to the two second-best methods Deep Ensembles and Rank-1 BNN, the gap between our methods increases when the level shift increases, showing our Density-Softmax is more robust than theirs under distributional shifts.*

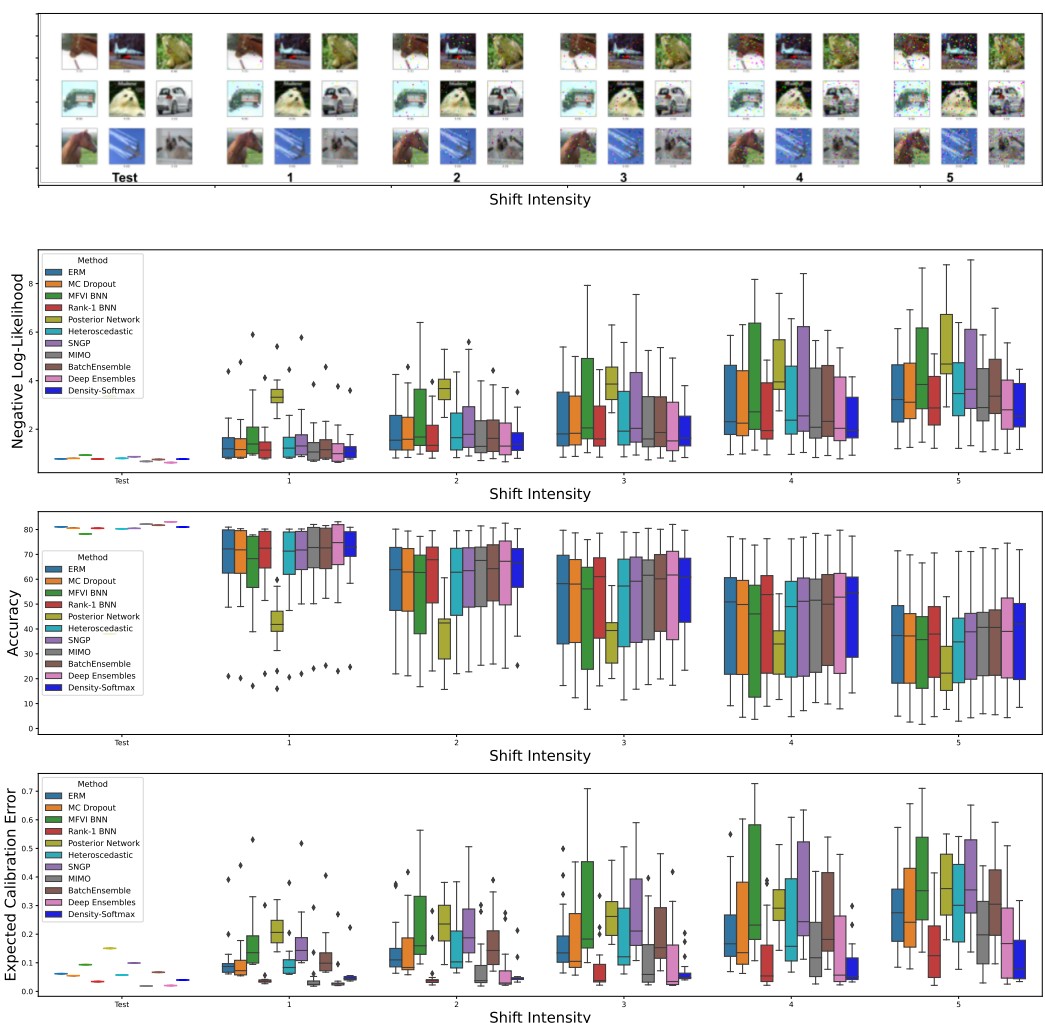

**Figure 14:** A corruption example by shift intensities and comparison under the distributional shift of negative log-likelihood, accuracy, and ECE under all types of corruptions in CIFAR-100-C. For each method, we show the mean on the test set and summarize the results on each shift intensity with a box plot. Each box shows the quartiles summarizing the results across all 17 types of shift while the error bars indicate the min and max across different shift types. **Our Density-Softmax achieves the highest accuracy, lowest NLL and ECE with a low variance across different shift intensities**.

Figure 14 however only shows the statistics across different shift methods. Therefore, we next analyze in detail the above results over 10 different random seeds. Specifically, we train each model with a specific random seed and then evaluate the result. We repeatedly do this evaluation 10 times with 10 different seed values. Finally, collect 10 of these results together and visualize the mean and standard deviation by the error bar for each shift intensity level. *Figure 15 shows these results in the CIFAR-10-C setting from Table 2, we observe that the result of our model and other baselines have a small variance, showing the consistency and stability across different random seeds.*

Similarly, Figure 16 is the comparison across 10 running with 10 different random seeds in terms of model storage and inference cost from Table 2. There is no variance in storage comparison since the model size is fixed and not affected by a random seed. Meanwhile, the latency variances across different seeds are minor since narrow error bars are in the inference cost comparison bar chart. *And this once again, confirms that the results are consistent and stable, our Density-Softmax is the fastest model with almost similar latency to a single Deterministic ERM.*

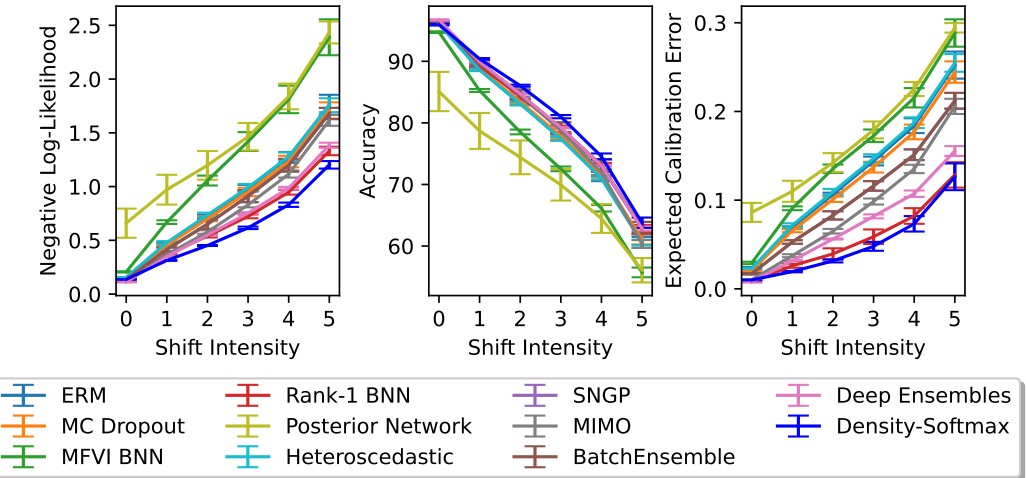

**Figure 15:** Benchmark performance on CIFAR-10-C with Wide Resnet-28-10 over 10 different random seeds. We plot NLL, accuracy, and ECE for varying corruption intensities; each result is the mean value over 10 runs and 15 corruption types. The error bars represent a fraction of the standard deviation across corruption types. **Our method achieves competitive results with SOTA with low variance across different shift intensities**.

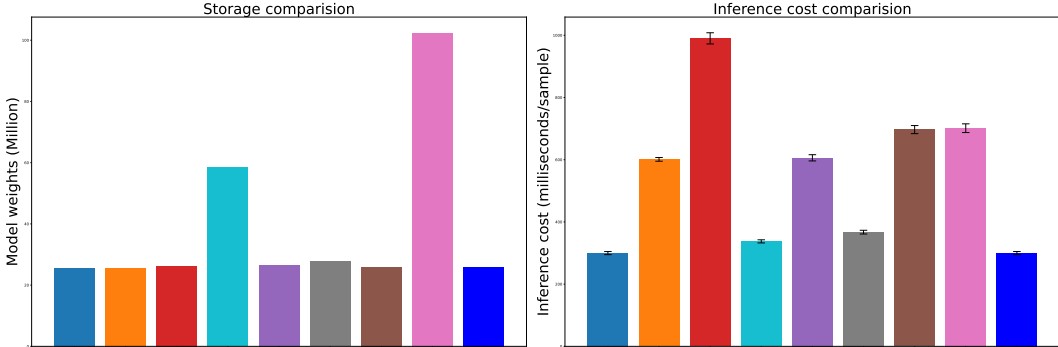

**Figure 16:** Comparison in model storage requirement (Million weights) and inference cost (milliseconds per sample) with error bars across 10 seeds of Resnet-50 on ImageNet dataset with NVIDIA RTX A5000. **Our Density-Softmax achieves almost the same weight and inference speed with a single Deterministic ERM model, as a result, outperforming other baselines in computational efficiency**.

### C.4.3 CALIBRATION DETAILS

To understand how models calibrate in more detail, we visualize the reliability diagrams to test the model's confidence across IID and OOD settings. For IID setting in CIFAR-10, Figure 17 illustrates our Density-Softmax is one of the best-calibrated models with a low ECE. On the one hand, compared to Deterministic ERM, MC Dropout, BatchEnsemble, and Deep Ensemble, our model is less under-confidence than them in the prediction that has lower than about $0.4$ confidence. On the other hand, compared to Deterministic ERM, MC Dropout, MFVI BNN, SNGP, and BatchEnsemble, our model is less over-confident than them in the prediction that has higher than about $0.4$ confidence. *As a result, accompanying Rank-1 BNN and Deep Ensembles, our model achieves the best calibration performance on IID data.*

More importantly, we find that Density-Softmax is calibrated better than other methods on OOD data in Figure 18 and Figure 19. In particular, Figure 18 shows our model and Deep Ensembles achieves the lowest ECE and less over-confident than others in a particular CIFAR-10-C test set. *Similarly, in real-world distributional shifts CIFAR-10.1 (v6), our model even achieves the lowest ECE with $0.01$, outperforms the SOTA Deep Ensembles. These results once again confirm the hypothesis that our model is one of the best reliable models and especially robust under distributional shifts.*

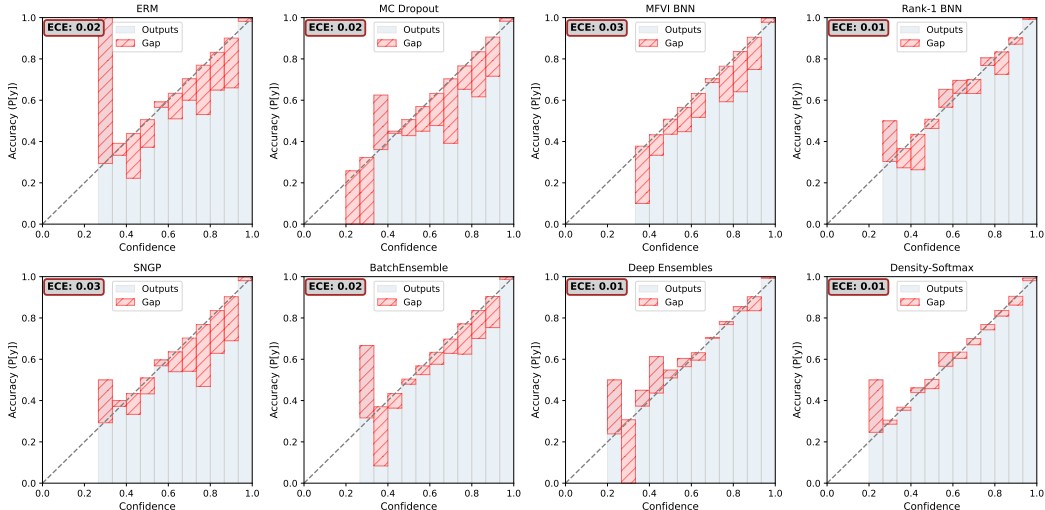

**Figure 17:** Reliability diagram of Density-Softmax versus different approaches, trained on CIFAR-10, test on IID CIFAR-10. **Density-Softmax has a competitive calibration result with SOTA methods**.

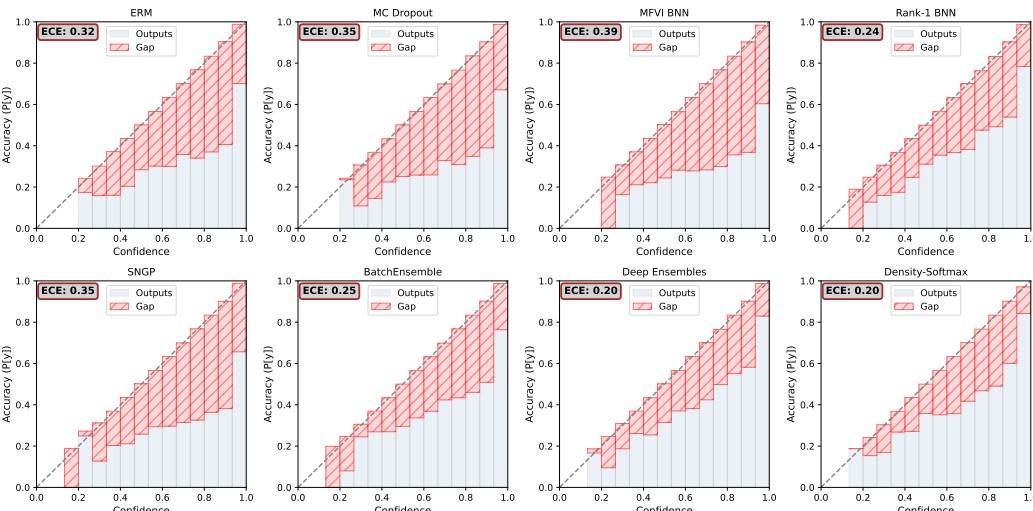

**Figure 18:** Reliability diagram of Density-Softmax versus different approaches, trained on CIFAR-10, test on OOD CIFAR-10-C set with "frosted glass blur" skew and "3" intensity. **Density-Softmax has a competitive calibration result with SOTA methods**.

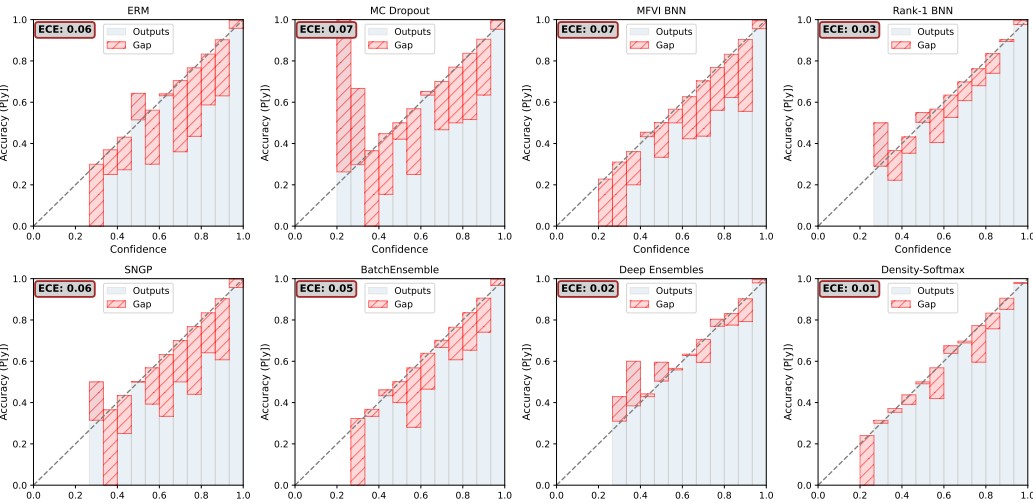

**Figure 19:** Reliability diagram of Density-Softmax versus different approaches, trained on CIFAR-10, test on real-world shifted OOD CIFAR-10.1 (v6). **Density-Softmax is better-calibrated than other methods**.

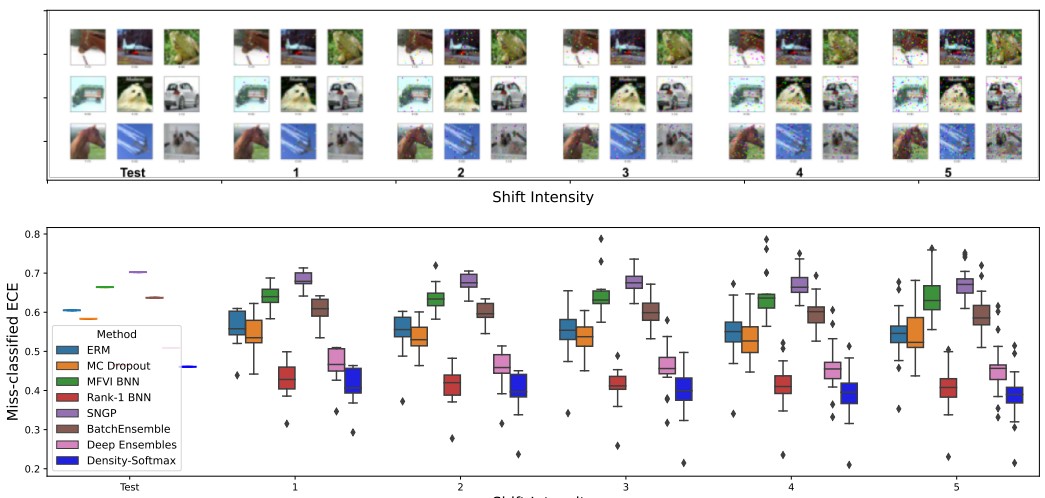

**Figure 20:** A corruption example by shift intensities and comparison under the distributional shift of miss-classified ECE under all types of corruptions in CIFAR-100-C (setting is similar to Figure 14). **Our Density-Softmax achieves the lowest miss-classified ECE with low variance across different shift intensities**.

**Miss-classified Expected Calibration Error.** To evaluate the calibration quality in more detail, we continue to make a comparison in terms of Miss-classified ECE (Chen et al., 2022). This measurement is a specific case of ECE in Eq. 40. It is different by covering only the miss-classified samples. In this case, the lower mECE, the more honest of show uncertainty if the model actually makes wrong predictions. Fig. 20 illustrates the box plots of mECE across shift intensities. *It confirms Density-Softmax achieves the best performance with the lowest mECE, showing the ability to say "I don't know" before it makes the wrong predictions.*

### C.4.4   PREDICTIVE ENTROPY DETAILS

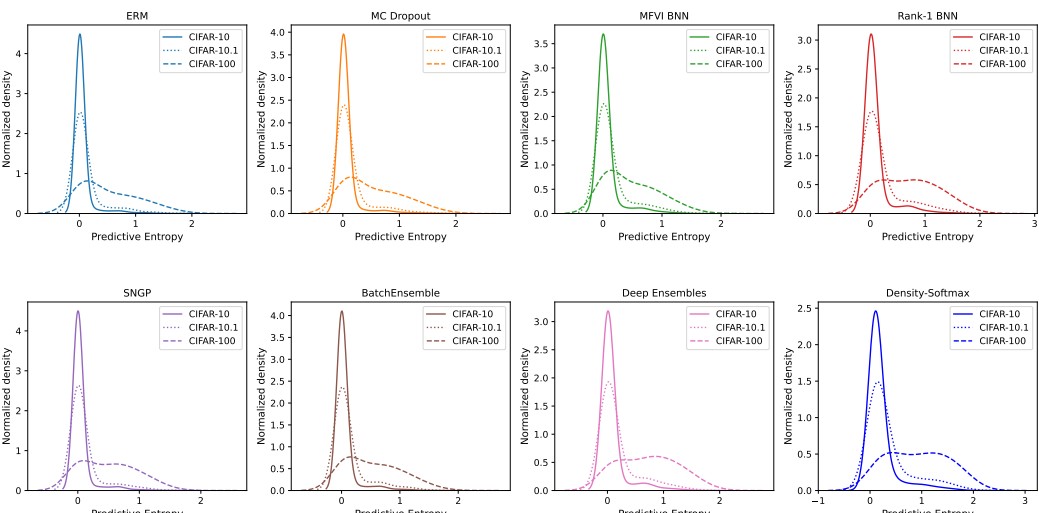

**Figure 21:** Histogram (bandwidth = 0.5) with density plot details from Figure 4 of predictive entropy for each method on IID testing data CIFAR-10 (solid lines), covariate shift with OOD CIFAR-10.1 (v6) (dotted lines), and semantic shift with completely different OOD CIFAR-100 (dashed lines). **Our Density-Softmax has a low predictive entropy value on IID while achieves the highest entropy value on OOD data**.

Fig. 4 has shown our trained model achieves the highest entropy value with a high density. Since this is the semantic shift, the highest entropy value indicates that our model is one of the best OOD detection models. Yet, this is only a necessary condition for a high-quality uncertainty estimation model because a pessimistic model could also achieve this performance. Therefore, to prove that Density-Softmax is not always under-confidence, we plot Fig. 21. We observe that Density-Softmax has a low predictive entropy with a high density on IID testing data, proving that our model is not under-confidence on IID data. *Combined with the under-confidence on OOD data results, these show our Density-Softmax is a reliability model in terms of uncertainty estimation.*

