# OpenReview forum: "Density-Softmax: Efficient Test-time Model for Uncertainty Estimation and Robustness under Distribution Shifts"
_ICLR.cc/2024/Conference — Submitted to ICLR 2024_

### Official Review · Reviewer_6MCn · 2023-10-19

**Soundness:** 1 poor
**Presentation:** 2 fair
**Contribution:** 2 fair
**Rating:** 3
**Confidence:** 4

**Summary:**

This paper proposes a method titled “density-softmax”, which aims at improving uncertainty estimation and robustness to distribution shifts of models, while being computationally efficient. The canonical methods in this space are Bayesian neural nets and ensembles, which are both computationally expensive, thus motivating this method. The proposed approach involves first building a Lipschitz-constrained model and then training normalizing flows on the feature descriptors. The data probabilities from the normalizing flows are combined with the softmax probabilities to yield “density softmax”, which is shown to minimize minimax uncertainty risk. Experiments show that the proposed approach is competitive with the computationally expensive counterparts.

**Strengths:**

- The proposed approach is simple to implement and seems to approximately match the performance of deep ensembles across accuracy, NLL, ECE across natural, corrupted and OOD data.
- The provided theory attempts to nicely illustrate some salient features of the method, namely, that it produces the correct probabilities when the test data is far away from the train data (theorem 4.3). Although I discuss some issues below.

**Weaknesses:**

**The role of the Lipschitz constraint is not conceptually clear; The resulting feature extractor may not even be 1-Lipschitz**

(1)The method incorporates two components: a Lipschitz-constrained feature extractor and the density-softmax. Experiments (Sec 5.5) show that the Lipschitzness helps OOD generalization, while the density softmax helps in uncertainty estimation. These components are largely independent of each other conceptually, as presented in the paper. This makes it difficult to answer the question - is Lipschitness truly a necessary component of the solution? Or is the introduction of a mere robustness constraint (via $\ell_2$ adversarial training, say) sufficient? Note that while 1-Lipschitz models are robust, robust models are not necessarily Lipschitz.

(2)It is also unclear whether the proposed training results in a 1-Lipschitz model or feature extractor. Note that training a model with a gradient norm penalty is insufficient to train a 1-Lipschitz model, as mere regularization does not imply that the constraint is satisfied. To verify whether models achieve the 1-Lipschitz guarantee, the method either needs to use provable methods such as [1] or computationally verify the existence of a small Lipschitz constant using [2].

---

**Missing comparison/discussion of a crucial baseline: joint probability estimation p(y,x)**


One alternate and simple way to incorporate knowledge of the input distribution is to estimate the joint probability $p(y, x) = p(y \mid x) p(x)$ [3]. This can be done by simply multiplying the softmax probabilities with the density estimates from the normalizing flow in the case of this paper. Note that this estimate satisfies all the asymptotic properties that density softmax does. Given the conceptual simplicity of this approach, I am wondering whether the authors can comment on the similarities/differences between their approach and joint probability estimation.

---

**Incorrect statements in Theorem 4.3; Possible flaws in proof of minimax optimality (corollary 4.4)**

(1)Theorem 4.3 shows the asymptotic behavior of density softmax. The statement claims that “if $d(z_{iid}, Z) \rightarrow 0$ then $p(z_{iid}, \alpha) \rightarrow 1$”. However, this is clearly untrue, as the maximum value of distribution itself can be less than one.

Consider for example the simple case of fitting a single multivariate Gaussian (i.e., a GMM with number of mixtures = 1) on data $Z$. Then clearly the max value of this distribution depends on the determinant of the covariance matrix, which can be any value, and in particular less than one if the determinant is large enough.

(2)Outside of this scaling issue, consider that the in-distribution is sampled from a Gaussian mixture model with a number of mixtures = 2, with probabilities 0.1 and 0.9 respectively for the two mixture components. Thus $p(z_{iid} (mixture 2)) \sim 9 \times p(z_{iid} (mixture 1))$. In other words, the normalizing flow cannot assign equal probability to all input samples unless the input distribution is uniform.

(3)Apart from the fact that the corollary builds on top of this possibly flawed theorem, it also makes the assumption that asymptotic conclusions derived for certain points $d(z_{iid},Z) \rightarrow 0$ can be applied to **every** IID point. This is simply incorrect. There can exist IID points that simply haven’t been sampled from the distribution yet and thus do not have near zero distances from the train set points.

---

[1] Singla et al., Improved techniques for deterministic l2 robustness, 2022

[2] Scaman et al., Lipschitz regularity of deep neural networks:
analysis and efficient estimation, 2018

[3] Grathwohl et al., Your classifier is secretly an energy based model and you should treat it like one, 2020

**Questions:**

Please address the points raised in the weaknesses section

---

> ### Author Response · Authors · 2023-11-17
> **Official Response to Reviewer 6MCn**
>
> Thanks for your valuable feedback. We additionally address your concerns below.
>
> **1. The role of the Lipschitz constraint is not clear; The resulting feature extractor $f$ may not be 1-Lipschitz**. As discussed in Apd.B.3, we note that the Lipschitz constraint is a necessary component to combine with the density function in our framework. Because it not only helps to improve robustness but also supports improving uncertainty performance by ablation results in Tab.5.
>
> Regarding the 1-Lipschitz, we claim that our feature extractor $f$ aims to satisfy 1-Lipschitz by gradient-penalty regularization. We do not claim that $f$ is 1-Lipschitz. We agree that our feature extractor $f$ may not be 1-Lipschitz. We will remove the sentence "1-Lipschitz feature extractor" to avoid misleading. However, we note that our feature extractor $f$ satisfy $\sup_{x}||\nabla_x f(x)||_2 = 1$ in training data with the Rademacher Theorem.
>
> Since $f$ is a neural network, we believe that there is no way to find the exact $L(f)$ at the moment. Even SeqLip [2] is only an approximation estimation method. SeqLip also mentioned that their upper bounds estimation is still extremely large for modern neural networks and probably overestimates the true $L(f)$. As requested by the reviewer, we additionally provide the upper bound of $L(f)$ by using SeqLip's algorithm with the Wide Renet 28-10 backbone on the CIFAR-10 dataset. We observe that the upper bounds of $L(f)$ are about $4.5 \times 10^6$ without gradient-penalty and $2.8 \times 10^4$ with gradient-penalty regularization. This does not show whether our true $L(f)$ is equal to 1 or not, but at least, it shows the gradient-penalty regularization can reduce the upper bound of $L(f)$. Regarding [1], this is only provable for a specific backbone, it is not true for the ResNet backbone we used in our experiments.
>
> **2. Comparison/discussion with joint probability estimation $p(y,x)$**. Thanks for your suggestion. However, we are slightly unsure of what you mean by this point. Any further clarification would be appreciated.
>
> Our goal is to improve the uncertainty quality of $p(y|x)$ by using the density model on feature space $p(z)$. Our motivation is not to use it for generative modeling so we only compare our method with models directly modeling the conditional probability. We also cannot formulate $p(y,x) = p(y|x) p(x)$, where $p(y|x)$ is softmax output and $p(x)$ is normalizing flow's output. Because we estimate feature space $p(z)$ not sample space $p(x)$.
>
> In terms of modeling $p(y|x)$, we agree with the reviewer that we can compare Density-Softmax with JEM [3]. However, according to Tab.1 in JEM, $p(y|x)$ of JEM only achieves $92.9\%$, and has a lower classification accuracy than the discriminative Derterministic-ERM ($95.8\%$) on CIFAR-10. As also mentioned by JEM, this worse performance is due to the fact that JEM is a hybrid model and needs to jointly optimize $p(y|x)$ with $\log p(x)$. Additionally, training $\log p(x)$ with SGLD is very sensitive and needs a long time to converge. In contrast, Density-Softmax is a discriminative model, archives a higher accuracy than Derterministic-ERM (see our Tab.2-3-4), and does not require any sampling during training.
>
> **3.1. Theorem 4.3: the maximum value of distribution itself can be less than one**. We agree with the reviewer that the maximum value of an arbitrary function $p(z; \alpha)$ can be less than 1. However, as discussed in Apd.B.1, we can re-scale the maximum output of $p(z; \alpha)$ to $1$ by dividing it by the maximum likelihood value in the training set. We also empirically show that the maximum value of $p(z; \alpha)$ equals $1$ with IID data in Fig.7 in our paper.
>
> **3.2. The normalizing flow cannot assign equal probability to all IID samples**. We do not claim that $p(z; \alpha)$ assigns the same likelihood for all IID samples. Please note that in Theorem 4.3, we only consider the ***strong*** IID input, i.e., when $d(z_{iid},Z_s)\rightarrow 0$.
>
> **3.3. Possible flaws in proof of minimax optimality (corollary 4.4)**. We agree with the reviewer that the statement "Density-Softmax satisfy by Theorem 4.3 and Proof A.1 for IID data" in our proof in Apd.A.2 does not apply to every IID data. We will remove this sentence. However, the result "$\inf_{\mathbb{P}}\sup_{\mathbb{P}^*}\left[S_{iid}(\mathbb{P},\mathbb{P^*})\right]$ is fixed" still holds due to Density-Softmax model’s predictive distribution learned from IID data (see condition (a) and proof in Proposition 2 in [4]). Please note that our density function does not modify the order of entries in the logit vector.
>
> We hope we have thoroughly addressed your concerns. We are willing to answer any additional questions.
>
> [1] Singla et al., Improved techniques..., 2022
>
> [2] Scaman et al., Lipschitz regula..., 2018
>
> [3] Grathwohl et al., Your clas..., 2020
>
> [4] Jeremiah et al., Simple and principled uncertainty estimation with deterministic deep learning via distance awareness, 2020

---

> > ### Comment · Reviewer_6MCn · 2023-11-21
> >
> > I have read the rebuttal and maintain my original score.

---

> > > ### Author Response · Authors · 2023-11-23
> > >
> > > Thanks for your response. If possible, we would like to know what are the specific remaining concerns after reading our rebuttal.

---

### Official Review · Reviewer_tUCH · 2023-10-23

**Soundness:** 4 excellent
**Presentation:** 4 excellent
**Contribution:** 2 fair
**Rating:** 5
**Confidence:** 4

**Summary:**

The paper introduces an approach to uncertainty quantification for deterministic neural networks. In contrast to Deep Ensembles and Bayesian methods, it does not require multiple forward passes of the data to produce an estimate of uncertainty. The method uses a density model (namely a normalizing flow), which scales predicted logits.

Thus, for the representations of new objects, that have a low density under this density model (typically it would be out-of-distribution objects), we will have uniform predictions.

The method has almost no additional computational overhead, which makes it attractive for practical usage.

**Strengths:**

The paper is very well written.

I found it very easy to follow.

I really like the amount of experiments and ablation studies conducted.

**Weaknesses:**

Major concerns:

1) My major concern is the novelty and overall contribution to the topic.

The same idea of using a density model (e.g. Normalizing flow) to scale  (by multiplication) some predicted parameters was used in NatPN [1]. I also wonder why, taking into account this similarity, the paper was not even discussed. It indeed was cited several times, but it was not discussed what is the gist of the method, and it was not compared.

The authors of the paper under review compared only with PostNet [2]. But there is a difference in parametrization between NatPN [1] and PostNet [2]. In [2] they learned a flow per class (which appeared to be redundant) and in the follow-up (and generalized) paper they introduced another parametrization, which needs only one flow (or another density model) overall. This explains why Latency for Posterior Net [2] scales with the number of classes (which should not be the case for NatPN [1]).
Moreover, in the problem of [1], they want to predict parameters of Dirichlet distribution, which is helpful as it allows them to distinguish between high aleatoric/epistemic uncertainties naturally. The parametrization of these predicted parameters of Dirichlet was in the form of the product of the density model and "update" to Dirichlet parameters. In the paper under review, authors do the same, but directly scale logits.

So it seems to me that it is pretty same idea, the same computational overhead (train one density model). The only difference is that in [1] they did the trick for the "second-order" distribution (which is Dirichlet), but here for the "first-order" distribution (Categorical), which does not have apparent benefits.


2) Normalization of the p(z; \alpha) to be in [0, 1]. I find it pretty strange and unnatural. First, why it is even an issue? How is it possible to have extremely large p(z;\alpha), given well trained Lipschitz feature extractor? Since we train the density model on top of the fixed extracted features (and well separated), it is surprising it is a problem.
Second, in the proof of Proposition 4.7, the fact that p(z; \alpha) is in [0, 1] is used. I wonder if will it be possible to prove it given p(z; \alpha) is still restricted, but say in [0, C], where C > 1.


3) Using the Lipschitz network is also not something new. I wonder why this option to satisfy the Lipschitz constraint was chosen (gradient penalty)?
In the Future Work there is a phrase "We plan to .... developing new techniques to avoid computing Jacobian matrix at training-time..." -- why iterative spectral normalization method (based on power method) [3] is not used for this purpose? It was applied in SNGP for example, in a paper cited by authors.


Minor concerns:


1) In the introduction: "However, it often struggles with over-confidence and overfitting (Guo et al., 2017). This poor performance **usually occurs when the test data is far and does not come from the same distribution as the training set**" -- I am a bit skeptical about it, as in [4] they observed this issue without shift when P_train(x, y) != P_test(x, y).
2) In the introduction: "recent sampling-based approaches..." but cited papers from 2016 and 2018.
3) In the introduction: I think the reference to Deep Ensembles should be [5].
4) In Proposition 4.7, why only an overconfident case is considered?
5) I did not find, what type of flow is used in the experiments?



[1] - Charpentier B. et al. Natural Posterior Network: Deep Bayesian Uncertainty for Exponential Family Distributions //arXiv preprint arXiv:2105.04471. – 2021.

[2] Charpentier B., Zügner D., Günnemann S. Posterior network: Uncertainty estimation without ood samples via density-based pseudo-counts //Advances in Neural Information Processing Systems. – 2020. – Т. 33. – С. 1356-1367.

[3] Miyato T. et al. Spectral normalization for generative adversarial networks //arXiv preprint arXiv:1802.05957. – 2018.

[4] Guo C. et al. On calibration of modern neural networks //International conference on machine learning. – PMLR, 2017. – С. 1321-1330.

[5] Lakshminarayanan B., Pritzel A., Blundell C. Simple and scalable predictive uncertainty estimation using deep ensembles //Advances in neural information processing systems. – 2017. – Т. 30.

**Questions:**

Please, address my concerns.

---

> ### Author Response · Authors · 2023-11-17
> **Official Response to Reviewer tUCH**
>
> Thanks for your valuable feedback. We additionally address your concerns below.
>
> **1. Comparison with NatPN**. We agree with the reviewer that we should add NatPN [1] to our baselines and we added them in our revised version. That said, NatPN does not consider improving the robustness and according to Tab.2-3 in NatPN [1], it even has a lower classification accuracy than Posterior~Net [2]. Additionally, different from our deterministic approach, NatPN is a conjugate-prior-based Bayesian Inference approach and requires predefined prior hyper-parameters for the model. This hyper-parameter is often sensitive and hard to choose in practice, which has been shown in the [uncertainty and robustness baseline codebase with Posteror-Net](https://github.com/google/uncertainty-baselines/tree/main/baselines/cifar). In contrast, our model does not require any pre-defined prior hyperparameter. Therefore, from the tables in our general response, we can see that our method is much more robust and also has a better uncertainty quality than NatPN.
>
> **2. Normalization of the $p(z; \alpha)$ to be in $[0, 1]$**. As discussed in Apd.B.1, we can re-scale the output of $p(z; \alpha)$ to be in $[0, 1]$ by dividing by the maximum likelihood value in the training set. We also empirically show this in Fig.7 in our paper. Regarding the proof of Proposition 4.7, if $c>1$, then Proposition 4.7 will not be true because the inequality in Eq. (46) does not hold.
>
> **3. Why the gradient penalty option to satisfy the Lipschitz constraint was chosen? Why Spectral Normalization is not used?**. Firstly, Spectral Normalization needs to use the power iteration method to find the eigenvalues of every weight in the model. This is more computationally expensive because it requires non-trainable vectors in the iterative process (see our Tab.2-3-4). Secondly, we did try Spectral Normalization and gradient-penalty regularization, and we found that the gradient-penalty regularization is better in terms of robustness. These observations are similar to the gap between Density-Softmax versus SNGP in terms of accuracy and latency in Tab.2-3-4 of our paper.
>
> **4. Minor Comments**. Thanks for your valuable suggestions. We will fix them in our revised version. Regarding the citation in our claim about distribution shifts, please note that we cite [4] and [5] to support our claim. Regarding Proposition 4.7, we only consider overconfidence because this is a major problem of current DNNs under distribution shifts. Regarding flow used in the experiments, we use radial flow and Real NVP.
>
> We hope we have thoroughly addressed your concerns. We are willing to answer any additional questions.
>
> [1] Bertrand Charpentier, Oliver Borchert, Daniel Z ugner, Simon Geisler, and Stephan Gunnemann. Natural posterior network: Deep bayesian predictive uncertainty for exponential family distributions. In International Conference on Learning Representations, 2022.
>
> [2] Bertrand Charpentier, Daniel Zugner, and Stephan Gunnemann. Posterior network: Uncertainty estimation without ood samples via density-based pseudo-counts. In Advances in Neural Information Processing Systems, 2020.
>
> [3] Jeremiah Liu, Zi Lin, Shreyas Padhy, Dustin Tran, Tania Bedrax Weiss, and Balaji Lakshminarayanan. Simple and principled uncertainty estimation with deterministic deep learning via distance awareness. In Advances in Neural Information Processing Systems, 2020.
>
> [4] Yaniv Ovadia, Emily Fertig, Jie Ren, Zachary Nado, D. Sculley, Sebastian Nowozin, Joshua Dillon, Balaji Lakshminarayanan, and Jasper Snoek. Can you trust your model's uncertainty? evaluating predictive uncertainty under dataset shift. In Advances in Neural Information Processing Systems, 2019.
>
> [5] Matthias Minderer, Josip Djolonga, Rob Romijnders, Frances Hubis, Xiaohua Zhai, Neil Houlsby, Dustin Tran, and Mario Lucic. Revisiting the calibration of modern neural networks. In Advances in Neural Information Processing Systems, 2021.

---

> > ### Comment · Reviewer_tUCH · 2023-11-17
> >
> > I would like to thank the authors for their detailed answers. I like that you conducted new experiments and added new results so quickly.
> >
> > However, my major concern was not properly addressed. I did not really ask to compare results with the NatPN but rather to discuss the differences/similarities with this paper.
> >
> > To be more precise, let's discuss formulas.
> >
> > Inference in NatPN (in the case of classification) effectively boils down to:
> >
> > $$
> > p(y \mid x, D) = \int_{\mu, \theta} p(y \mid \mu)p(\mu \mid x, \theta) p(\theta \mid D) d\mu d\theta \approx \int_{\mu} p(y \mid \mu)p(\mu \mid x, \hat{\theta}) d\mu,
> > $$
> >
> > where $p(\mu \mid x, \hat{\theta})$ is a Dirichlet distribution, and $p(y \mid \mu)$ is a Categorical one.
> >
> > Each $\mu$ is sampled from the simplex, parameterized by Dirichlet with parameters $\alpha(x) = \alpha_{prior} + p(f(x)) \tilde{g}(f(x)) = \alpha_{prior} + p(z) \tilde{g}(z)$, where $z=f(x)$, same as in author's approach, and $\tilde{g}$ is a classifier (vector of softmax values). Note, that both $\alpha(x)$ and $\alpha_{prior}$ are vectors of size $K$.
> >
> > For this pair of distributions, we know the closed-form result, which is:
> > $$
> > p(y \mid x, D) = \text{Cat}(\frac{\alpha(x)}{\sum_i \alpha_i(x)}),
> > $$
> > so the NatPN can be considered as a deterministic network in this sense as well: all the parameters of the network are deterministic, and the prediction requires only one forward pass as everything is in closed form.
> >
> > The choice of the prior is actually also pretty natural, $\alpha_{prior} = 1$, which leads to the flat distribution over the simplex. Thus, if there is a tiny density assigned to an input object $x$, then prior will dominate and the resulting prediction will be uniform, hence $p(y=c \mid x, D) = \frac{1}{K},$ where $K$ is the number of classes.
> >
> >
> > In your case, you do the same thing, however not for the update of parameters of Dirichlet, but rather directly for Categorical:
> >
> > $$
> > p(y \mid x, D) = \text{Cat}(\text{Softmax}[p(z)g(z)]),
> > $$
> > and in your case $g(z)$ is a vector of logits.
> >
> > Again, if there is no evidence to an input $x$, the prediction will be uniform.
> >
> > In your case, the measure of epistemic uncertainty is the $p(z)$. But the same measure, apart from the entropy of the resulting Dirichlet, can be used in NatPN. So everything is pretty much the same as in the author's paper.
> >
> >
> > So the main idea of both methods is to use this scaling of predictions by density, learned from some "representations" of the training data. The trick is simply applied in another place. That is why I raised my concern regarding novelty.
> >
> > Given these similarities, I think it is worth discussing 1) what are the ideological similarities/dissimilarities of these two approaches, 2) what are the benefits of the proposed approach, and 3) why NatPN is performing so poorly, given that it consists of the same components.

---

> ### Author Response · Authors · 2023-11-18
> **Follow-up Response to Reviewer tUCH**
>
> We gratefully appreciate your prompt response to our rebuttal. We discuss the similarities and differences of our method with NatPN as follows.
>
> **1. Similarities**. We agree with the reviewer that both of us have the same idea in terms of: (1) Estimating a marginal density model on feature space. (2) Combining this density model with the classifier to improve the model's uncertainty performance. (3) From the Bayesian view, both of us can be considered as a point estimate approach.
>
> **2. Differences**. To easily compare, following the reviewer's notation, let us derive as follows. Since $\tilde{g}$ denotes the vector of softmax, we have the Categorial distribution from the closed-form of Nat-PN is
>
> $$p(y|x,D) = Cat\left(\frac{\alpha(x)}{\sum_i \alpha_i(x)}\right) = Cat\left(\frac{\alpha_{\text{prior}} + p(z) \tilde{g}(z)}{\sum_i \alpha_{\text{ith-prior}} + p(z) \tilde{g}(z)_{i}}\right)$$
>
> $$ = Cat\left(\frac{\alpha_{\text{prior}} + p(z) \frac{\exp(g(z))}{\sum_{k=1}^K \exp(g(z)_k)}}{\sum_i \alpha\_{\text{ith-prior}} + p(z) \frac{\exp(g(z)_i)}{\sum\_{k=1}^K\exp(g(z)_k)}}\right), \text{(1)}$$
>
> where $\frac{\exp(g(z)\_i)}{\sum\_{k=1}^K \exp(g(z)\_k)} \leq 1, \forall i \in [1,\cdots, K]$, $\alpha_{\text{ith-prior}}$ and $g(z)\_i$ represent the $i$-th entry in the prior vector $\alpha_{\text{prior}}$ and logit vector $g(z)$ respectively. And, the Categorial distribution of our Density-Softmax is
>
> \begin{align}
>     p(y|x,D) = Cat\left(\text{Softmax}\left[p(z)g(z)\right]\right) = Cat\left(\frac{\exp(p(z) g(z))}{\sum_{i=1}^K \exp(p(z) g(z)_k)}\right). \text{(2)}
> \end{align}
>
> From the equation, we have the following key differences:
>
> - (1) From the Bayesian view, optimizing with Eq.1 is equivalent to doing Maximum a posteriori estimation (MAP) while optimizing with Eq.2 is equivalent to doing Maximum likelihood estimation (MLE).
>
> - (2) Eq.1 shows that the $p(y|x,D)$ of NatPN is not a standard softmax output because it is not normalized by a natural exponent function. Meanwhile, Eq.2 is normalized by a natural exponent function.
>
> **3. Benefits of our approach**. From the difference (1) and (2) above, we have the benefit of Density-Softmax over NatPN as follows:
>
> - (1) Eq.2 shows that Density-Softmax can calculate $p(y|x,D)$ without the need to consider how to define $\alpha(x)$. Please note that the $\alpha(x)$ can hurt $p(y|x,D)$ performance if we pre-define a bad prior.
>
> - (2) The natural exponential in Eq.2 helps Density-Softmax easily optimize with the cross-entropy loss by the nice derivative property of $\ln(\exp(a)) = a$, and produce a sharper distribution with a higher probability of the largest entry in the logit vector $g(z)$ and lower probabilities of the smaller entries when compared with the normalization of NatPN in Eq.1.
>
> **4. Why NatPN can have a bad performance**. Given the two benefits of Density-softmax mentioned, let us give two examples to show why NatPN can potentially have a bad performance:
>
> - (1) Let's consider we have three classes. As mentioned by the reviewer, NatPN will pre-define $\alpha_{\text{prior}} \sim \mathbb{U}$, where $\mathbb{U}$ stands for a uniform distribution. Plugin to Eq.1, we can see that if we choose $\alpha_{\text{ith-prior}} \gg p(z)$, then $\alpha_{\text{ith-prior}} \gg p(z)\frac{\exp(g(z)\_i)}{\sum_{k=1}^K \exp(g(z)_k)}$, yielding $p(y|x,D) \rightarrow \mathbb{U}$, i.e., $p(y|x,D) = (0.33,0.33,0.33)$. Therefore, NatPN's accuracy can be low and the predictive distribution is not sharp in this classification task.
>
> - (2) Now consider a sample $x$ with the true label $y=(1,0,0)$, $z=f(x)$, $g(z) = (4,2,2)$, $p(z) = 1$, $\alpha_{\text{prior}} = (1,1,1)$, then plug into Eq.(1), NatPN prediction is $\hat{y}_1=(0.4468, 0.2766, 0.2766)$. Meanwhile, from Eq.(2), Density-Softmax prediction is $\hat{y}_2=(0.7870, 0.1065, 0.1065)$. We can see that from $\hat{y}_1$ of NatPN, the probability of the largest entry in $g(z)$ is lower, and the probability of lower entries is higher when compared to $\hat{y}_2$. This potentially leads to a more flat distribution and results in lower accuracy and higher ECE for correct predictions.
>
> We hope we have thoroughly addressed your concerns. We are willing to answer any additional questions.

---

> > ### Comment · Reviewer_tUCH · 2023-11-20
> >
> > Dear authors,
> >
> > Thank you for your comments!
> >
> > 1) I think that this discussion should be explicitly incorporated into the paper because, given the similarities of the ideas in NatPN and the paper under review, readers should have a chance to fairly evaluate the contribution of the proposed approach.
> >
> > 2) About the sections "Benefits of our approach" and "Why NatPN can have a bad performance":
> >
> > - 1. While I agree that a bad choice of prior can hurt the performance of the method, I am still not persuaded that $\alpha_{prior} = 1$ is a bad one.
> >
> > If we make Maclaurin series for exponent (for small $p(z)$) in case of Density-Softmax, we will result in something like $\exp{[p(z)g(z)]} \sim 1 + p(z)g(z)$, which will be almost equivalent to what we have for $p(y \mid x, D)$ in NatPN for $\alpha_{prior} = 1$ with the only difference that $g(z)$ in your case are logits, but not vector of probabilities. But anyway it should not change the argmax in this case. So for small values of the density, the prediction (argmax) ideally should be the same.
> >
> > - 2. Having $\log (\exp (a))$ far it is also not really a convincing benefit. Maybe it makes the optimization slightly more robust, but I don't have a clear intuition about the effect.
> >
> >
> > Given the discussion we have, I am willing to slightly increase the score. Still, my concern about novelty is there.

---

> ### Author Response · Authors · 2023-11-21
> **Follow-up Response to Reviewer tUCH**
>
> Thanks for your further questions. As suggested, we added a detailed discussion about the comparison with NatPN in Appendix B.4 in our revised version. We will add more detail following our further discussions in the next version. We would like to continue to resolve your remaining concerns as follows.
>
> **1. The selection of $\alpha_{\text{prior}} = 1$**. We agree with the reviewer that if $p(z)$ is small enough, then $\arg\max$ between NatPN and Density-Softmax are the same at test-time. However, in training time, since optimizing in Eq.(1) with NatPN equivalents to do MAP, we can see that the predictive distribution of the posterior can be biased by the prior information. Therefore, if we select $\alpha_{\text{ith-prior}} = 1, \forall i \in \\{1,\cdots, K\\}$, i.e., $\alpha_{\text{prior}} \sim \mathbb{U}$, then the lower $p(z)$, the flatter of $p(y|x,D)$, and if $p(z)\ll 1$, then $p(y|x,D) \rightarrow \mathbb{U}$. We think this potentially makes $p(y|x,D)$ of NatPN under-fitting because its predictive distribution is flatter and not as sharp as Density-Softmax. Please note that Density-Softmax is trained by MLE and without $p(z)$ information in our pre-training step. In fine-tuning step, we fix $p(z)$ and only optimize with the classifier weight.
>
> **2. Benefits of $\log(\exp(a))$ in training**.  In this regard, let us follow the explanation in 6.2.Gradient-Based Learning, 6.2.2.Output Units, and 6.2.2.3.Softmax Units for Multinoulli Output Distributions, of the Deep Learning book [1]. As we mentioned in the different (1) in our previous response, the optimization in Eq.(2) with Density-Softmax equivalents to MLE, i.e., minimizing the negative log-likelihood, equivalently described as the cross-entropy between the training data and the model distribution as follows
>
> $$\min_{\theta}\\{-\mathbb{E}\_{(x,y)\sim D} \log(p_{\theta}(y|x))\\} = \min_{\theta}\left\\{-\sum_{j=1}^N \log(\text{Softmax}(p(z)g_\theta(z)_i))\right\\},$$
>
> where $\theta$ is model parameter, $N$ is the number of training data, and $i$ is the correct class for sample $j$-th. Therefore, let $a = p(z)g_\theta(z)$, i.e., $\text{Softmax}(a)\_i = \frac{\exp(a_i)}{\sum_{k=1}^K\exp(a_k)}$, then the log-likelihood can undo the $\exp$ of the softmax as
>
> $$\log(\text{Softmax}(a)\_i) = \log\left(\frac{\exp(a_i)}{\sum_{k=1}^K\exp(a_k)}\right) = \log(\exp(a_i)) - \log\left(\sum_{k=1}^K\exp(a_k)\right) = a_i - \log\left(\sum_{k=1}^K\exp(a_k)\right). \text{(3)}$$
>
> From Eq.(3), we can see that the $\exp$ in Density-Softmax roughly cancels out the $\log$ in the cross-entropy loss causing the loss to be roughly linear in $a_i$, i.e., if the correct answer already has the largest input to the softmax, then the $-a_i$ term and the $\log(\sum_{k=1}^K \exp(a_k)) \approx \max_{k \in \\{1,\cdots,K\\}} a_k = a_i$ terms will roughly cancel. This leads to a roughly constant gradient and the correct sample contributes little to the overall training cost, which will be dominated by other examples that are not yet correctly classified, allowing the model to correct itself quickly in training [1]. Therefore, a wrong saturated Density-softmax does not cause a vanishing gradient and its predictive distribution is potentially sharp in training [1].
>
> We hope we have thoroughly addressed your concerns. We are willing to answer any additional questions.
>
> [1] Deep Learning (Ian J. Goodfellow, Yoshua Bengio and Aaron Courville), MIT Press, 2016.

---

### Official Review · Reviewer_ueNw · 2023-10-28

**Soundness:** 3 good
**Presentation:** 3 good
**Contribution:** 3 good
**Rating:** 8
**Confidence:** 5

**Summary:**

This paper is about uncertainty estimation with a single model, the authors propose an architecture that combines feature density estimation with a normalizing flow into the softmax activation at the output layer, in a way that density combined with the logits going to the softmax allow for much less overconfident predictions.

Contributions are:
- The Density-Softmax layer/model, where a single model combines feature density estimation with the softmax activation at the output for improved uncertainty estimation.
- Theoretical proofs that the proposed model is solution to the minimax uncertainty risk and is distance aware in feature space and significantly reduces overconfidence of softmax for out of distribution inputs.
- Experimental results show that the proposed method is an advance of the field, with comparable accuracy and improved calibration and out of distribution detection capabilities on CIFAR-10/100, ImageNet, and corrupted versions of them, by only requiring a single forward pass of the network.

**Strengths:**

- The paper is well written and clear to understand, minus some minor details I mention below.
- The topic of single network uncertainties is important and has broad application in the field. One large reason why methods for uncertainty estimation in machine learning models are not used in practice, is because of their increased computation costs, and single network models with uncertainty can produce proper calibrated uncertainties without significant additional computation required.
- The field of uncertainty from distance awareness in feature spaces has advanced considerably in the last years, and yet still feature space confidences were separate from softmax probabilities, this paper proposes to integrate the feature density as estimated by a normalizing flow into the softmax activation, by basically scaling the logits with the feature density, which seem to work well to reduce softmax overconfidence.
- The evaluation and experimental setup seems to be high quality, minus the selection of baselines as I comment in weaknesses, there is a good selection of datasets (CIFAR10/100, ImageNet, and their corrupted versions), with state of the art convolutional networks, and a good selection of metrics covering task performance, uncertainty quality and calibration, and out of distribution detection.
- Experimental results show that the proposed method outperforms the baselines in terms of uncertainty quality, and produces similar results for task performance (accuracy). The proposed method has some troubles outperforming deep ensembles, but this is understandable as a single model is used while ensembles have much larger complexity.
- The experimental evaluation is very comprehensive, it covers task performance on multiple datasets and networks, uncertainty quality from multiple aspects, including calibration error for in and out of distribution datasets (with corrupted versions of datasets), and quality of feature density estimation and its relation to out of distribution examples.

**Weaknesses:**

~~- I believe the largest weakness in this paper are the baselines, while there are many baselines covering recent methods from the state of the art, some recent single network methods are missing and these are direct competitors of Density-Softmax:~~

~~Van Amersfoort, Joost, et al. "Uncertainty estimation using a single deep deterministic neural network." International conference on machine learning. PMLR, 2020. This paper defines DUQ which uses a RBF layer instead of softmax, also being feature space distance aware.~~

~~Mukhoti, Jishnu, et al. "Deep Deterministic Uncertainty: A New Simple Baseline." Proceedings of the IEEE/CVF Conference on Computer Vision and Pattern Recognition. 2023. This paper defines DDU which uses 1-Lipschitz networks and feature space density as a signal for epistemic uncertainty, while softmax outputs are used for aleatoric uncertainty. The proposed method Density-Softmax is a direct competitor of DDU.~~

~~To me it is very clear that at least DUQ and DDU are clear baselines, specially DDU, as to me the only difference between Density-Softmax and DDU is the use of normalizing flows for density estimatin and combination of feature density into the softmax activation in the last layer.~~

~~- Also a notable point is that DDU outperforms most of the baselines used in this paper (like SNGP and ERM) and is very competitive with ensembles, producing better uncertainty than 5 ensembles in the out of distribution setting, and having slightly lower calibration error. This supports my proposal that Density-Softmax should be compared to DDU. DDU has been part of the public literature for around a year.~~

~~- One important concept that seems to be missing from this paper is aleatoric and epistemic uncertainty. Feature density estimates epistemic uncertainty (shown and proven in the DDU paper), while softmax probabilities model aleatoric uncertainty. Since the proposed model Density-Softmax modifies softmax probabilities, these now are more closely estimating epistemic uncertainty, and then, what component of the model outputs aleatoric uncertainty? This could be a possible disadvantage of this model.~~

Minor Comments

~~- The links to each abbreviation, highlighted in red, are a bit too much distraction, I suggest that for abbreviations, the authors turn off the links highlighted in red, the reader will thank you. I would even suggest to replace link boxes with colored links using hyperref options.~~
- In Figure 1, each plot contains the x and y axis labels, these do not really add any new information to the figure and I suggest to remove them, for cleanliness.
- Overall the paper focuses too much into mathematical details and proofs, I would suggest to add the intuition of Density-Softmax close to the beginning of the paper, the user only finds out how the method works in around page 4-5, and the intuition is simple, scale logits with the feature density, which conceptually is similar to temperature scaling with a variable temperature.

**Questions:**

- How do you compare your proposed method Density-Softmax with Deep Deterministic Uncertainty (DDU)?
- Could you comment on how aleatoric and epistemic uncertainty are output by your model?

After the rebuttal, the authors have addressed my questions and weaknesses, particularly about comparison to DDU, and now I am satisfied that the paper can be accepted.

---

> ### Author Response · Authors · 2023-11-17
> **Official Response to Reviewer ueNw**
>
> Thanks for your valuable feedback. We additionally address your concerns below.
>
> **1. Comparison with DUQ**. From the tables in our general response, we can see that our method is much better than DUQ [1] in all criteria. Please note that our reported number is consistent with Tab.2-3 in SNGP [3].
>
> **2. Comparison with DDU**. Firstly, according to Tab.1 in DDU [2], DDU's ECE is $0.85$ and $4.10$, while Deep Ensembles is $0.76$ and $3.32$ on CIFAR-10 and CIFAR-100. So, their reported uncertainty performance is worse than Deep Ensembles in terms of calibration error by having a ***higher*** ECE. Secondly, DDU does not consider improving accuracy as their motivation, especially the robustness under distribution shifts. Finally, we want to emphasize that the DDU method uses the post-hoc re-calibration technique to optimize temperature variables to improve the uncertainty performance, while, our Denisty-Softmax does not use any re-calibration set. We believe that the number of samples in the re-calibration set for temperature scaling is a signification number to be considered. For example, DDU needs 5000 samples in the calibration set to optimize, while the training set has 45000 samples in CIFAR-10. Therefore, we only compare the performance of DDU without temperature scaling. From the figures, we can see that our method is also better than them in all criteria.
>
> **3. How aleatoric and epistemic uncertainty is output by your model?**. Similarly to DDU, we can easily disentangle aleatoric uncertainty by computing the softmax probability without multiplying with the density model by our Eq.(5) in the paper, and epistemic uncertainty by using the likelihood value of the density model.
>
> **4. Minor Comments**. Thanks for your valuable suggestions. We will fix them in our revised version.
>
> We hope we have thoroughly addressed your concerns. We are willing to answer any additional questions.
>
> [1] Joost Van Amersfoort, Lewis Smith, Yee Whye Teh, and Yarin Gal. Uncertainty estimation using a single deep deterministic neural network. In Proceedings of the 37th International Conference on Machine Learning, 2020.
>
> [2] Jishnu Mukhoti, Andreas Kirsch, Joost van Amersfoort, Philip HS Torr, and Yarin Gal. Deep deterministic uncertainty: A new simple baseline. In Proceedings of the IEEE/CVF Conference on Computer Vision and Pattern Recognition, 2023.
>
> [3] Jeremiah Liu, Zi Lin, Shreyas Padhy, Dustin Tran, Tania Bedrax Weiss, and Balaji Lakshminarayanan. Simple and principled uncertainty estimation with deterministic deep learning via distance awareness. In Advances in Neural Information Processing Systems, 2020.

---

> > ### Comment · Reviewer_ueNw · 2023-11-23
> > **Response to rebuttal**
> >
> > Dear Authors,
> > Thank you for the additional information and answers, I believe this answers my questions and concerns, specially about baselines and uncertainty disentanglement,  I will be updating my review and increasing my score.
> >
> > Just one small detail, in the latency results in the table above, why is the DUQ latency similar to ensembles? It is a single network model so it should have good latency.

---

### Official Review · Reviewer_ZCiU · 2023-11-04

**Soundness:** 3 good
**Presentation:** 2 fair
**Contribution:** 2 fair
**Rating:** 5
**Confidence:** 3

**Summary:**

In this paper, the authors present a framework for uncertainty quantification by combining the classification with density estimation to identify any test-time distribution shifts. This allows estimating model confidence with one single forward pass, as compared to most BNN approaches or ensemble methods that require multiple forward passes at test time. The authors present results comparing their approach with other existing approaches on the image classification task (also involving test-time distribution shifts)

**Strengths:**

- Popular methods for uncertainty quantification, such as deep ensembles, involve multiple forward passes (at least in a naive implementation) and as a result can be computationally prohibitive. I like that this paper proposes a single-pass approach.

- The overall approach is very intuitive - having a density estimation to alter the model confidence is an understandable strategy.

I also applaud the fact that the authors have made their implementation available publicly and anonymously.

**Weaknesses:**

- The main weakness of this paper is that it doesn't look into other recent methods proposed - including those that involve direct uncertainty estimation [1-3], and distillation of BNNs/ensembles [4, 5].

- As the authors have noted, it seems that the proposed approach would have a higher training time and higher memory requirements due to the Jacobian matrix in the regularization term.

References

[1] Lahlou, S., Jain, M., Nekoei, H., Butoi, V. I., Bertin, P., Rector-Brooks, J., ... & Bengio, Y. (2022). DEUP: Direct Epistemic Uncertainty Prediction. Transactions on Machine Learning Research.

[2] Zhang, J., Dai, Y., Xiang, M., Fan, D. P., Moghadam, P., He, M., ... & Barnes, N. (2021). Dense uncertainty estimation. arXiv preprint arXiv:2110.06427.

[3] van Amersfoort, J., Smith, L., Jesson, A., Key, O., & Gal, Y. (2021). On feature collapse and deep kernel learning for single forward pass uncertainty. arXiv preprint arXiv:2102.11409.

[4] Vadera, M., Jalaian, B., & Marlin, B. (2020, August). Generalized bayesian posterior expectation distillation for deep neural networks. In Conference on Uncertainty in Artificial Intelligence (pp. 719-728). PMLR.

[5] Malinin, A., Mlodozeniec, B., & Gales, M. (2019, September). Ensemble Distribution Distillation. In International Conference on Learning Representations.

**Questions:**

- Rather than just multiplying the density term with the classifier outputs, what would be the impact of passing them through a parameterized model (such as MLP)? That would add more flexibility to the overall approach and improve performance, I believe.

- Can the authors comment on how their approach would compare with other distillation approaches highlighted earlier?

- Do the authors have some insights on how the entire algorithm can be made faster?

---

> ### Author Response · Authors · 2023-11-17
> **Official Response to Reviewer ZCiU**
>
> Thanks for your valuable feedback. We additionally address your concerns below.
>
> **1. Comparison with [1-3]**. [1-2] are sampling-based methods which suffer from a worse efficiency performance. Additionally, according to page 15 in DEUP [1], DEUP only archives 93.89% accuracy on CIFAR-10, which is lower than our method (96%) and Deep Ensembles (96.6%). Please note that DEUP also uses the same [uncertainty and robustness baseline codebase](https://github.com/google/uncertainty-baselines/tree/main/baselines/cifar). Meanwhile, [2] is only a technical report, has not been accepted to any conference, and also does not publish source code. DUE [3] can be a sampling-free method and we add them for comparison in our revised version. From the tables in our general response, we observe that our method outperforms DUE in all criteria. It is also worth noting that DUE [3] has not been accepted to a main conference yet.
>
> **2. What if we pass the density term with the classifier outputs through a parameterized model?**. Our motivation is to use the density function to reduce the over-confidence of the DNN classifier outputs under distribution shifts. If we pass the density term with the classifier outputs through an MLP model, the accuracy of the parameterized MLP model can be different from the classifier outputs, and the over-confidence issue of the classifier outputs may not be resolved. In contrast, if we multiply the density term with the classifier outputs, the model's accuracy still holds. Therefore, we can show in Prop.4.7. that our method can significantly mitigate the over-confidence issue of the classifier.
>
> **3. How Density-Softmax compare with other distillation approaches in [4-5]?**. Thank you for your suggestion. We are aware that distillation approaches are trying to also speed up the inference steps. We have cited them and added more discussion about these distillation approaches in our revised version. However, in the distillation approaches, [4-5] has shown the student model still underperforms the teacher model BNNs and Deep Ensembles in terms of uncertainty and robustness. Given that we directly compare our method with BNNs and Deep Ensembles and show that we outperform BNNs and have a competitive result with Deep Ensembles, it implies our model should be better than distillation approaches (e.g., [4-5]) in terms of uncertainty quality and robustness.
>
> **4. Do the authors have some insights on how the entire algorithm can be made faster?**. We are slightly unsure whether you mean faster training or testing performance. Regarding testing performance, as mentioned in Remark 3.4, our algorithm is sampling-free by requiring only a single forward pass to make an inference. Therefore, our method is faster than other sampling-based baselines. Regarding training performance, as discussed in our future work, we expect to develop new techniques to avoid computing the Jacobian matrix to improve training time performance in future work.
>
> We hope we have thoroughly addressed your concerns. We are willing to answer any additional questions.
>
> [1] Lahlou, S., Jain, M., Nekoei, H., Butoi, V. I., Bertin, P., Rector-Brooks, J., ... \& Bengio, Y. (2022). DEUP: Direct Epistemic Uncertainty Prediction. Transactions on Machine Learning Research.
>
> [2] Zhang, J., Dai, Y., Xiang, M., Fan, D. P., Moghadam, P., He, M., ... \& Barnes, N. (2021). Dense uncertainty estimation. arXiv preprint arXiv:2110.06427.
>
> [3] van Amersfoort, J., Smith, L., Jesson, A., Key, O., \& Gal, Y. (2021). On feature collapse and deep kernel learning for single forward pass uncertainty. arXiv preprint arXiv:2102.11409.
>
> [4] Vadera, M., Jalaian, B., \& Marlin, B. (2020, August). Generalized bayesian posterior expectation distillation for deep neural networks. In Conference on Uncertainty in Artificial Intelligence (pp. 719-728). PMLR.
>
> [5] Malinin, A., Mlodozeniec, B., \& Gales, M. (2019, September). Ensemble Distribution Distillation. In International Conference on Learning Representations.

---

> > ### Comment · Reviewer_ZCiU · 2023-11-22
> > **Response to the rebuttal**
> >
> > I thank the authors for engaging in the discussion. It's good to see the additional results mentioned by the authors.
> >
> > - On point 1, the preprint by Zhang et al., 2021 [1], does publish their code. I found their GitHub repo in the abstract here: https://arxiv.org/pdf/2110.06427.pdf (github: https://github.com/JingZhang617/UncertaintyEstimation). I agree that it doesn't seem to be published, but having an open-source implementation makes me curious. I won't be considering this as a major weakness.
> > - On point 2, I guess it'll be useful to actually run this experiment and see it. The nice thing about MLP is that theoretically, it can still learn the multiplication function (among all the other functions), and some scaled versions of it too.
> > - Fair response on point 3 - no more concerns here. But you should include the discussion and references in the paper as well (not sure if already done).
> > - On point 4, I was alluding to improving training time. If the authors have some ideas since they wrote the rebuttal, it'll be useful to know.
> >
> >
> > References
> >
> > [1] Zhang, J., Dai, Y., Xiang, M., Fan, D. P., Moghadam, P., He, M., ... & Barnes, N. (2021). Dense uncertainty estimation. arXiv preprint arXiv:2110.06427.

---

### Author Response · Authors · 2023-11-17
**Additional results with DUQ, DDU (without Temperature Scaling), DUE, and NatPN on CIFAR-100**

| **Method**          | **NLL($\downarrow$)** | **Acc($\uparrow$)** | **ECE($\downarrow$)** | **cNLL($\downarrow$)** | **cAcc($\uparrow$)** | **cECE($\downarrow$)** | **AUPR-S($\uparrow$)** | **AUPR-C($\uparrow$)** | **\#Params($\downarrow$)** | **Latency($\downarrow$)** |
|--------------------------|----------------------------|--------------------------|----------------------------|-----------------------------|---------------------------|-----------------------------|-----------------------------|-----------------------------|---------------------------------|--------------------------------|
| DUQ                      | 0.980                      | 78.5                     | 0.119                      | 2.84                        | 50.4                      | 0.281                       | 0.878                       | 0.732                       | 77.58M                          | 1,547.35                       |
| DDU (w/o TS)             | 0.877                      | 79.7                     | 0.086                      | 2.70                        | 51.3                      | 0.240                       | 0.890                       | 0.797                       | 37.60M                          | 959.25                       |
| DUE                      | 0.902                      | 79.1                     | 0.038                      | 2.32                        | 53.5                      | 0.127                       | 0.897                       | 0.787                       | 37.50M                          | 926.99                         |
| NatPN                    | 1.249                      | 76.9                     | 0.091                      | 2.97                        | 48.0                      | 0.265                       | 0.875                       | 0.768                       | 36.64M                          | 613.44                         |
| Deep Ensembles           | **0.666**             | **82.7**            | 0.021                      | 2.27                        | 54.1                      | 0.138                       | 0.888                       | 0.780                       | 146.22M                         | 1,569.23                       |
| **Density-Softmax** | 0.780                      | 80.8                     | 0.038                      | **1.96**               | **54.7**             | **0.089**              | 0.910                       | **0.804**              | **36.64M**            | **522.94**  |

---

> ### Author Response · Authors · 2023-11-21
> **Waiting for your response before the discussion period ends**
>
> Dear reviewers,
>
> We would like to thank you again for spending your time reading the rebuttals as well as evaluating our paper. We truly appreciate that.
>
> As the end of the discussion period is approaching, we look forward to hearing your feedback if our answers haven’t addressed your questions. We would be happy to discuss if you still have any concerns. If our answer resolves your concerns, we kindly ask whether the score can be raised.
>
> We did update our revised paper following your valuable feedback. We hope we have thoroughly addressed your concerns.
>
> All the best,
>
> Authors.

---

> > ### Author Response · Authors · 2023-11-23
> > **Update our PDF revision**
> >
> > We thank the reviewers again for helping us improve our paper. We did upload and update our revised version following the reviewer’s feedback and our discussion so far. We would like to list what we updated (colored by \textcolor{blue}{blue}) in our new version as follows:
> > - We added more discussion about our comparison with related works, especially NatPN in Appendix B.4.
> > - We added the comparison tables with DUQ, DDU, DUE, and NatPN in Tables 6 and 7.
> > - We changed the phrase “1-Lipschitz feature extractor” to “Lipschitz-constrained feature extractor” and removed the sentence “Density-Softmax satisfy by Theorem 4.3 and Proof A.1 for IID data” in our proof in Apd.A.2 to avoid misleading.
> > - We turned off the links highlighted in red and changed the link colors to avoid distraction.
> >
> > Given the discussion period is ending soon, it would be nice to know whether you have remaining concerns after reading our rebuttals. This definitely will help us to improve the quality of the paper in the future.

---

### Author Response · Authors · 2023-11-17
**Additional results with DUQ, DDU (without Temperature Scaling), DUE, and NatPN on CIFAR-10**

| **Method**          | **NLL($\downarrow$)** | **Acc($\uparrow$)** | **ECE($\downarrow$)** | **cNLL($\downarrow$)** | **cAcc($\uparrow$)** | **cECE($\downarrow$)** | **oNLL($\downarrow$)** | **oAcc($\uparrow$)** | **oECE($\downarrow$)** | **\#Params($\downarrow$)** | **Latency($\downarrow$)** |
|--------------------------|----------------------------|--------------------------|----------------------------|-----------------------------|---------------------------|-----------------------------|-----------------------------|---------------------------|-----------------------------|---------------------------------|--------------------------------|
| DUQ                      | 0.239                      | 94.7                     | 0.034                      | 1.35                        | 71.6                      | 0.183                       | 0.49                        | 87.9                      | 0.068                       | 40.61M                          | 1,538.35                       |
| DDU (w/o TS)             | 0.159                      | 96.0                     | 0.024                      | 1.06                        | 76.0                      | 0.153                       | 0.39                        | 89.8                      | 0.063                       | 37.60M                          | 954.31                       |
| DUE                      | 0.145                      | 95.6                     | 0.007                      | 0.84                        | 77.8                      | 0.079                       | 0.46                        | 89.2                      | 0.066                       | 37.50M                          | 916.26                         |
| NatPN                    | 0.242                      | 92.8                     | 0.041                      | 0.89                        | 73.9                      | 0.121                       | 0.46                        | 86.3                      | 0.049                       | 36.58M                          | 601.35                         |
| Deep Ensembles           | **0.114**             | **96.6**            | 0.010                      | 0.81                        | 77.9                      | 0.087                       | 0.28                        | **92.2**             | 0.025                       | 145.99M                         | 1,520.34                       |
| **Density-Softmax** | 0.137                      | 96.0                     | 0.010                      | **0.68**               | **79.2**             | **0.060**              | **0.26**               | 91.6                      | **0.016**              | **36.58M**            | **520.53**  |

---

### Author Response · Authors · 2023-11-17
**General Response**

We thank all reviewers for spending time reviewing our paper, along with constructive feedback. Since the major concerns of Reviewer ZCiU, ueNw, and tUCH are our novelty and missing comparison with related works, we would like to make a general response to address these concerns.

**1. Novelty**. Our paper proposes a theoretically sound framework to achieve three criteria: uncertainty quality, robustness, and test-time efficiency. To the best of our knowledge, our paper is the first work that efficiently combines a density function built on a Lipschitz-constrained feature extractor with the softmax layer to improve the three aforementioned criteria. According to the [uncertainty and robustness baseline codebase](https://github.com/google/uncertainty-baselines/tree/main), the best algorithm for uncertainty and robustness is still Deep Ensembles. However, Deep Ensembles is not efficient. We empirically show our algorithm archives a competitive result with Deep Ensembles while outperforming them in terms of efficiency. This implies that our Density-Softmax is among one of the best efficient models for uncertainty estimation and robustness.

**2. Comparison with DUQ, DDU, DUE, and NatPN**. As suggested by reviewers, we provide the detailed comparison below and we also added them in Tab.6-7 in our revised version. We observe that all of them underperform our method.

We also want to point out that we did not add these methods to our experiment in our initial version for three reasons. Firstly, no one of them considers improving accuracy and robustness under distribution shifts. Secondly, in terms of uncertainty quality, their reported results are also generally worse than Deep Ensembles [1-5].

Finally, we want to make several comments about specific methods mentioned by the reviewers:
- DDU [2] uses temperature scaling with an additional re-calibration set, while we focus on methods that do not require additional calibration data and post-hoc calibration steps.
- Regarding DUQ [1], according to Tab.2-3 in SNGP [5], SNGP outperforms DUQ in all criteria. Given the fact that our method outperforms SNGP, we can roughly conclude that we will also outperform DUQ. This is validated by the new results shown in the table.
- NatPN [4] is based on Posterior-Net [6], which has been reported to have significantly worse results than other baselines by the [uncertainty baselines codebase with Posterior-Net](https://github.com/google/uncertainty-baselines/tree/main/baselines/cifar). We are aware that there is a difference between the numbers reported in the paper Posterior-Net [6] and the uncertainty baselines. We believe the reason is that this Bayesian approach is sensitive to different model backbones by requiring predefined prior hyper-parameters. Given we are following the implementation of uncertainty baselines to ensure a fair comparison, we conclude that our method will outperform NatPN, which is also validated by the new experimental results in the table.

Last but not least, the reason we use uncertainty baselines codebase is that it provides the same configuration in terms of setup environments, model architectures, hyperparameters, and data preprocessing across methods. We believe that this is more fair for comparison among different methods.

**We appreciate the reviewer's feedback and we hope the reviewers can reconsider the given score after seeing our general response above and individual responses below.**

[1] Joost Van Amersfoort, Lewis Smith, Yee Whye Teh, and Yarin Gal. Uncertainty estimation using a single deep deterministic neural network. In Proceedings of the 37th International Conference on Machine Learning, 2020.

[2] Jishnu Mukhoti, Andreas Kirsch, Joost van Amersfoort, Philip HS Torr, and Yarin Gal. Deep deterministic uncertainty: A new simple baseline. In Proceedings of the IEEE/CVF Conference on Computer Vision and Pattern Recognition, 2023.

[3] Joost van Amersfoort, Lewis Smith, Andrew Jesson, Oscar Key, and Yarin Gal. On feature collapse and deep kernel learning for single forward pass uncertainty, 2022.

[4] Bertrand Charpentier, Oliver Borchert, Daniel Z ugner, Simon Geisler, and Stephan Gunnemann. Natural posterior network: Deep bayesian predictive uncertainty for exponential family distributions. In International Conference on Learning Representations, 2022.

[5] Jeremiah Liu, Zi Lin, Shreyas Padhy, Dustin Tran, Tania Bedrax Weiss, and Balaji Lakshminarayanan. Simple and principled uncertainty estimation with deterministic deep learning via distance awareness. In Advances in Neural Information Processing Systems, 2020.

[6] Bertrand Charpentier, Daniel Zugner, and Stephan Gunnemann. Posterior network: Uncertainty estimation without ood samples via density-based pseudo-counts. In Advances in Neural Information Processing Systems, 2020.

---

### Meta-Review · Area_Chair_Mutu · 2023-12-04

**Metareview:**

This paper presents a method for quantifying uncertainty of neural network without requiring sampling or ensembles. The proposed method brings together two components proposed in prior work: 1) a density estimate of the final extracted features (based on a normalizing flow) and 2) a regularization term that ensures that the extracted features are 1-Lipschitz. During the initial reviewing phase, there were concerns about novelty and a lack of baselines. Since this work is combining components that have been independently proposed in the literature, there is a higher “burden of proof” that their combination is a significant methodological contribution. This is to some degree accomplished by the authors’ theory, as well as the post-rebuttal experimental comparisons. Nevertheless, due to the significant amount of experiments and discussion that took place during the reviewing period, it is clear that this paper could benefit from an additional round of revisions before publications.

**Justification For Why Not Higher Score:**

All of the reviewers had concerns about novelty and comparison to baselines. Though the authors address these concerns in the rebuttal, including adding new experimental results, the amount of discussion and proposed changes merits an additional round of reviewing before publication.

**Justification For Why Not Lower Score:**

N/A

---

### Decision · Program_Chairs · 2024-01-16

Reject